# ONE MODEL FOR ALL: MULTI-OBJECTIVE CONTROLLABLE LANGUAGE MODELS

## ABSTRACT

Aligning large language models (LLMs) with human preference is critical to enhancing LLMs' safety, helpfulness, helpfulness, humor, faithfulness, etc. The current reinforcement learning from human feedback (RLHF) mainly focuses on a fixed reward learned from average human ratings, which may weaken the adaptivity and controllability of varying preferences. However, creating personalized LLMs requires aligning LLMs with individual human preferences, which is non-trivial due to the scarce data per user and the diversity of user preferences on multi-objective trade-offs, such as prioritizing humor and empathy in one context, while seeking efficiency and precision in another. *Can we train one LLM to produce personalized outputs for different user preferences on the Pareto front?* In this paper, we introduce Multi-Objective Control (MOC), which trains an LLM as a meta-policy to directly generate responses in the preference-defined regions of Pareto front. Our approach integrates multi-objective optimization (MOO) principles into Proximal Policy Optimization (PPO) to train an LLM as a preference-conditioned policy network. We improve the computational efficiency of MOC by applying MOO at the policy level, which enables us to finetune an LLM of 7B parameters on a single A6000 GPU. Extensive experiments demonstrate the advantages of MOC over baselines in three aspects: (i) Controllability of LLM outputs w.r.t. user preferences on the trade-off among multiple rewards; (ii) Quality and diversity of LLM outputs, measured by the hyper-volume of multiple solutions achieved; and (iii) Generalization to unseen preferences. These results highlight MOC's potential for real-world applications requiring scalable and customizable LLMs.

## 1 INTRODUCTION

Large language models (LLMs) have gained significant attention for their impressive performance across a wide range of tasks, including machine translation (Vaswani et al., 2017; Radford & Narasimhan, 2018; Devlin et al., 2019), text generation (Touvron et al., 2023; OpenAI, 2023), and conversational agents (Ouyang et al., 2022; Bai et al., 2022). However, these models are generally aligned with fixed preferences predetermined by developers (Ouyang et al., 2022; Touvron et al., 2023; Bai et al., 2023; Dubey et al., 2024), limiting the available degree of personalization to the users. In real-world scenarios, users often have diverse preferences for LLMs behavior. For instance, one user might prefer a humorous and empathetic response for emotional support, while another might prioritize a more efficient, task-oriented assistant. Despite this variability, the inherent flexibility of current LLMs (Dubey et al., 2024; OpenAI, 2023) is limited to provide fully personalized interactions.

The ability of LLMs to adjust their behavior according to diverse user preferences is called *multi-objective controllability*, a crucial feature for enhancing user satisfaction. Multi-objective controllability allows a model to dynamically balance the trade-offs between different objectives based on user-defined preferences. Training separate models for each preference order, however, is neither practical nor scalable due to the high computational costs. That highlights the need to enable one-time LLM training while accommodating a broad range of preferences.

Can we **control** the trade-offs in a single, once-trained LLMs to meet diverse human preferences? Our answer is **yes**. This paper aims to (i) enable LLMs to generate customized responses for diverse user preferences and (ii) achieve this with a once-trained model. To this end, we introduce a novel algorithm, Multi-Objective Control (MOC), which leverages a carefully designed multi-

objective optimization (MOO) algorithm. MOC requires only one training, incorporates explicit policy improvement, and does not rely on human preference data. Moreover, its training cost is comparable to single-objective reinforcement learning from human feedback method (RLHF) (Schulman et al., 2017; Ouyang et al., 2022), and we made it feasible to fine-tune a 7-billion model on a single A6000 GPU with LoRA (Hu et al., 2022).

We first formulate the multi-objective controllability as an MOO problem with preference vector constraints, inspired by recent advancements in MOO (Désidéri, 2009; Sener & Koltun, 2018; Xiao et al., 2023). This formulation presents two primary challenges. The first one is identifying the target to be controlled. Existing MOO works typically focus on optimizing different loss functions(Liu et al., 2021; 2023) or linearized utility functions (Yang et al., 2019), which do not effectively capture the quality or behavior of LLMs. In contrast, MOC selects the reward signal as the control target, enabling direct manipulation of the model's behavior. The second challenge is to solve this optimization problem within feasible computational limits. Our formulated optimization problem involves complex trade-offs among multiple objectives under different preference constraints. To address this, we relax the problem into a new form of MOO, where the preference constraint is treated as an additional objective. MOC scalarizes the objectives with dynamic weighting in different steps, ensuring the computational cost comparable to the widely used single-objective RLHF (Schulman et al., 2017; Ouyang et al., 2022). Table 1 provides a detailed comparison of MOC and baseline methods.

In extensive experiments, MOC consistently outperforms baseline methods (Ouyang et al., 2022; Ramé et al., 2023; Yang et al., 2024b) across multiple tasks. It demonstrates strong performance in three key areas: (i) Controllability, as it effectively aligns model behavior with diverse preference vectors and ensures a monotonic relationship between input preferences and outcomes; (ii) Solution set quality, measured by the hyper-volume metric, where MOC achieves a superior Pareto front while maintaining a diverse set of solutions; and (iii) Generalization, as it robustly handles unseen preferences while preserving both the alignment quality and diversity. Compared to baseline methods, MOC offers a more efficient and flexible approach to personalizing LLMs, managing different trade-offs among multiple objectives with a single model and seamlessly adapting to new preferences. These findings highlight MOC's potential for real-world applications requiring scalable and customizable personalization.

Our contributions are as follows: (i) We introduce the MOC algorithm, which takes comparable computation as single-objective RLHF and finetunes LLMs only once to accommodate diverse user preferences; (ii) We empirically validate MOC, demonstrating its superior performance in terms of controllability, solution quality, and generalization, including its ability to generalize to unseen user preferences.

Table 1: Comparison with the state-of-the-art MOO methods. MOC addresses MOO more principally and efficiently. $M-$ the number of preferences, $N-$ the number of reward models (objectives).

| Algorithms | Explicit policy improvement | Num of trained LLMs | Inference adaptation | Preference data | Loss |
|---|---|---|---|---|---|
| MORLHF | ✓ | $M$ | × | No | PPO |
| Rewarded Soups (Ramé et al., 2023) | × | $N$ | ✓ | No | PPO |
| MODPO (Zhou et al., 2024) | ✓ | $M$ | × | Yes | DPO |
| RiC (Yang et al., 2024b) | × | 1 | ✓ | No | SFT |
| MOC (Ours) | ✓ | 1 | ✓ | No | PPO |

## 2 BACKGROUND

**RLHF** (Ouyang et al., 2022) consists of reward modeling and policy optimization phases. The reward model is trained by maximizing $\mathcal{L}_{RM} = \mathbb{E}_{(x,y^w,y^l)\sim\mathcal{D}}[\log(\sigma(r(x,y^w) - r(x,y^l)))]$, where $y^w/y^l$ mark the wanted/unwanted response, $\sigma(\cdot)$ denotes the sigmoid function, and $x$ is the prompt. Typical RLHF leverages the Proximal Policy Optimization (Schulman et al., 2017) (PPO) for policy optimization: $\arg\max_{\pi(y|x;\theta)} \mathbb{E}_{x\sim\mathcal{D},y\sim\pi(\cdot|x)}[r(x,y) - \beta\log\frac{\pi(y|x;\theta)}{\pi_{old}(y|x)}]$.

**Controllability v.s. Alignment.** In this paper, we call a model *controllable* if it can inherently behave differently according to different user preferences, i.e., in line with the user's expectations. *Alignment* refers to the language model being aligned to a common preference (usually defined by the developer) which does not change.

## 3 MULTI-OBJECTIVE CONTROLLABLE LANGUAGE MODELS

Can we **control** the trade-offs in once-trained language models to accommodate diverse user preferences? Our answer is "**Yes**". Our goal is twofold: (i) Enabling the language model to satisfy a wide range of user preferences, and (ii) Achieving this with a model trained only once.

To represent user preferences, we define a vector $\mathbf{p} = [p_1, p_2, \cdots, p_N]$, $\sum_{i=1}^{N} p_i = 1, p_i \geq 0$, where each element in $\mathbf{p}$ reflects the importance of a specific objective. Inspired by recent work on multi-objective learning (Xu et al., 2020; Ma et al., 2020; Yang et al., 2022), we use this preference vector to regulate the model's output in the objective space. Given $M$ preference vectors $\{\mathbf{p}^i\}_{i=1}^{M}$, training a LLM controllable by the preference vectors is formulated as the following optimization problem:

$$\max_{\theta} \mathbf{J}(\pi(\cdot; \theta, \mathbf{p}^i)) \stackrel{\text{def}}{=} \max_{\theta}(J^1(\pi(\cdot; \theta, \mathbf{p}^i)), J^2(\pi(\cdot; \theta, \mathbf{p}^i)), \cdots, J^N(\pi(\cdot; \theta, \mathbf{p}^i)))^{\top},$$

$$\text{s.t. } \Phi\Big(\pi(\cdot; \theta, \mathbf{p}^i) \| \mathbf{p}^i\Big) \leq \phi, \forall i \in \{1, 2, 3, \cdots, M\}, \tag{1}$$

where $J^i$ denotes the RLHF objective associated with reward $R^i$. The LLM is a meta-policy $\pi$ parameterized by $\theta$ and takes a preference vector $\mathbf{p}$ as an input condition. In addition, $\Phi$ is a distance or divergence metric between the policy $\pi$ and the preference vector $\mathbf{p}$, and the controllability requires a distance upper bounded by $\phi$. Generally, the objective $J^i$ is selected as a PPO loss (Schulman et al., 2017; Ouyang et al., 2022). $J^i$ is next all selected as PPO loss unless specified.

Conventional approaches to solving constrained optimization problems, such as the Lagrangian method, are inefficient for handling the complexity of Equation (1) due to the multiple constraints, diverse preferences, and the high dimensionality of language model parameters. This insufficiency renders developing new solutions imperative.

### 3.1 WHAT SHOULD THE PREFERENCE VECTOR ALIGN WITH?

Existing multi-objective learning methods (Yang et al., 2019; Liu et al., 2023; 2021) typically focus on balancing multiple loss functions. However, RL loss is not necessarily an indicator of the agent's performance and thus is not suitable as the target of control. In contrast, the value function or episodic return is a better performance measure. In RLHF of LLMs, the reward is evaluated by a reward model and serves as the episodic return. Therefore, we choose a multi-dimensional reward signal as the primary target for control. To maintain simplicity, we select mean squared error (MSE) as the similarity metric between the reward signal and the preference vector. Formally, the constraint in Equation (1) is specified as

$$\Phi\Big(\pi(\cdot; \theta, \mathbf{p}^i) \| \mathbf{p}^i\Big) \stackrel{\text{def}}{=} MSE\Big(\mathbb{E}_{x \sim \mathcal{D}} \mathbf{R}(x, y), \mathbf{p}^i\Big) \leq \phi, \tag{2}$$

where $x$ represents the prompt/query, $y \sim \pi(x; \theta, \mathbf{p}^i)$ is LLM-generated response, and $\mathcal{D}$ is the prompt dataset. The reward vector $\mathbf{R}(x, y) = (R^1(x, y), R^2(x, y), \cdots, R^N(x, y))$ is associated with the $N$ optimization objectives $\{J^i\}_{i=1}^{N}$. The sampled response $y$ depends on the policy parameters $\theta$, which allows optimization of $\mathbf{J}(\pi)$ with respect to $\theta$ through standard RLHF. Equation (2) enforces that the reward vector $\mathbf{R}(x, y)$ aligns closely with the preference vector $\mathbf{p}^i$. In other contexts such as typical RL settings, the value function can be the target of control. Further details are provided in Appendix C.

**Re-labelling the prompt.** The meta-policy $\pi$ in MOC takes an extra condition on a preference vector $\mathbf{p} = [p_1, p_2, \cdots, p_N]$. Hence, we modify the original prompt by appending the preference vector to it, i.e.,

$$\text{Re-labeled prompt = <R1> } p_1 \text{ <R2> } p_2 \text{ ... <RN> } p_N \text{ \{prompt\}.} \tag{3}$$

### 3.2 MULTI-OBJECTIVE ALIGNMENT OF LLMS

To solve the multi-objective learning problem with inequality constraints, we introduce our Multi-Objective Control (MOC) algorithm, which builds on recent advances of multi-objective learning (Désidéri, 2009; Sener & Koltun, 2018). The MOC simultaneously optimizes all the objectives while maximizing the similarity between the objective value vector and the preference vector. We

optimize the following similarity objective due to its simplicity:

$$\max_{\theta} J^{\Phi} \stackrel{\text{def}}{=} \max_{\theta} -ReLU(MSE(\mathbb{E}_{x \sim \mathcal{D}}\mathbf{R}(x, y), \mathbf{p}^j) - \phi), \tag{4}$$

where $ReLU(x) = \max(x, 0)$ penalizes constraint violations when the error exceeds the threshold $\phi$. This ensures that optimization respects the trade-offs between rewards and preferences. The gradient of Equation (4) can be approximated as

$$\nabla_{\theta} ReLU(MSE(\mathbb{E}_{x \sim \mathcal{D}}\mathbf{R}(x, y), \mathbf{p}^j) - \phi) = \mathbf{1}_{MSE(\mathbb{E}_{x \sim \mathcal{D}}\mathbf{R}(x,y), \mathbf{p}^j) - \phi > 0} \sum_{k=1}^{N} (R^k - p_k^j) \nabla_{\theta} R^k(x, y), \tag{5}$$

where $\mathbf{1}_{(\cdot)}$ is the indicator function, $R^k$ represents the $k^{th}$ entry of $\mathbf{R}$, $p_k^j$ means the $k^{th}$ entry of preference vector $p^j$. Besides, $\nabla_{\theta} R^k(x, y)$ aims at maximizing the corresponding rewards, which is also the gradient of the PPO loss aim at. Thus, one could use the PPO objective $\nabla_{\theta} J^k(\pi(\cdot; \theta, \mathbf{p}^j)$ to compute $\nabla_{\theta} R^k(x, y)$.

Solving the original optimization problem in Equation (2) is computationally challenging because it involves $N$ objectives and $M$ preferences. Thus, we reformulate it as

$$\max_{\theta} \widehat{\mathbf{J}}(\pi(\cdot; \theta, \mathbf{p}^i)) \stackrel{\text{def}}{=} \max_{\theta} \left( \mathbf{p}^{i^{\top}} \mathbf{J}(\pi(\cdot; \theta, \mathbf{p}^i)), -ReLU(MSE(\mathbb{E}_{x \sim \mathcal{D}}\mathbf{R}(x, y), \mathbf{p}^i) - \phi) \right)^{\top}, \tag{6}$$

where $\mathbf{J}(\pi(\cdot; \theta, \mathbf{p}^i))$ is defined in Equation (2). This reformulation offers two significant advantages: (i) It significantly reduces optimization complexity by transforming the original $N$-objective optimization into a bi-objective optimization; (ii) It retains the control over the preference vectors in the newly formulated optimization problem. Scalarization simplifies the problem even further:

$$\max_{\theta} \left\{ c^{(1)} \mathbf{p}^{i^{\top}} \mathbf{J}(\pi(\cdot; \theta, \mathbf{p}^i)) - c^{(2)} ReLU(MSE(\mathbb{E}_{x \sim \mathcal{D}}\mathbf{R}(x, y), \mathbf{p}^i) - \phi) \Big| \sum_{i=1}^{2} c^{(i)} = 1, c^{(i)} \geq 0 \right\}, \tag{7}$$

where $c^{(i)}$ is an $i$-objective related co-efficient, determined by solving a min-norm problem

$$\min_{c^{(1)}, c^{(2)}} \left\{ \left\| c^{(1)} \mathbf{p}^{i^{\top}} \nabla_{\theta} \mathbf{J}(\pi(\cdot; \theta, \mathbf{p}^i)) - c^{(2)} \nabla_{\theta} ReLU(MSE(\mathbb{E}_{x \sim \mathcal{D}}\mathbf{R}(x, y), \mathbf{p}^i) - \phi) \right\|_2^2 \Big| \sum_{i=1}^{2} c^{(i)} = 1, c^{(i)} \geq 0 \right\}. \tag{8}$$

As demonstrated by Désidéri (2009); Sener & Koltun (2018), either: (i) The solution to this min-norm problem is zero, in which case the resulting point satisfies the KKT conditions; or (ii) The solution yields a gradient direction that improves all objectives.

### 3.3 MULTI-OBJECTIVE ALIGNMENT OF LLMS AT SCALE WITH SURROGATE

However, in the context of LLMs, directly addressing this optimization remains intractable in computation because: (i) the need to backpropagate $N + 1$ times to compute the gradient for each objective; and (ii) solving the min-norm problem in the gradient space for LLM parameters is prohibitively expensive in computation. To overcome this computational burden, we introduce a more efficient-to-optimize surrogate, which is an upper bound to the original objective, circumventing the need for costly backpropagation operations.

**Theorem 1.** *The upper bound of Equation* (8) *is*

$$\left\| c^{(1)} \sum_{j=1}^{N} p_j^i I(\hat{A}_j) - c^{(2)} \mathbf{1}_{MSE(\mathbb{E}_{x \sim \mathcal{D}}\mathbf{R}(x,y), \mathbf{p}^i) - \phi > 0} \sum_{j=1}^{N} (R^j - p_j^i) I(\hat{A}_j) \right\|_2^2 \left\| \nabla_{\theta} \pi(\cdot; \theta, \mathbf{p}^i) \right\|_2^2, \tag{9}$$

*where*

$$I(A) = \begin{cases} 0, & \text{if } (A > 0 \text{ and } z > (1 + \epsilon)) \text{ or } (A < 0 \text{ and } z < 1 - \epsilon) \\ A, & \text{if } (A > 0 \text{ and } z \leq (1 + \epsilon)) \text{ or } (A < 0 \text{ and } z \geq 1 - \epsilon) \end{cases}; \tag{10}$$

$$\sum_{i=1}^{2} c^{(i)} = 1, \quad c^{(i)} \geq 0 \quad \forall i; \tag{11}$$

*the advantage function $A$, the clip hyper-parameter $\epsilon$, and the ratio $z = \frac{\pi}{\pi_{old}}$ are introduced by the PPO loss (Schulman et al., 2017).*

The proof is deferred to Appendix A. Theorem 1 provides an upper bound on Equation (8), which yields two key advantages: (i) Both $I(\hat{A}_i)$ and $\mathbf{1}_{MSE(\mathbb{E}_{x \sim \mathcal{D}} \mathbf{R}(x,y), \mathbf{p}^i) - \phi > 0} \sum_{j=1}^{N} (R^j - p_j^i) I(\hat{A}_j)$ can be efficiently computed without any additional expensive back-propagation; (ii) $\nabla_\theta \pi(\cdot; \theta, \mathbf{p}^i)$ is no longer required by the min-norm problem since it does not depend on $c^{(i)}$. Therefore, we achieve the following computationally efficient surrogate problem of optimizing $c^{(1)}$ and $c^{(2)}$:

$$\min_{c^{(i)}} \left\{ \left\| c^{(1)} \sum_{j=1}^{N} p_j^i I(\hat{A}_j) - c^{(2)} \mathbf{1}_{MSE(\mathbb{E}_{x \sim \mathcal{D}} \mathbf{R}(x,y), \mathbf{p}^i) - \phi > 0} \sum_{j=1}^{N} (R^j - p_j^i) I(\hat{A}_j) \right\|_2^2 \middle| \sum_{i=1}^{N} c^{(i)} = 1, c^{(i)} \geq 0, \forall i \right\}. \tag{12}$$

Compared to the intractable original optimization in Equation (8), the surrogate optimization problem in Equation (12) offers the following advantages: (i) Computational efficiency: The term $I(\hat{A}_i)$ can be computed through a simple forward pass in a language model without requiring gradient calculations; (ii) Solution efficiency: Note that the objective function is a quadratic function of the variables $c^{(i)}$. The general min-norm problem is solvable by the existing Frank-Wolfe algorithm (Jaggi, 2013), a well-established convex optimization method. Equation (12) has a closed-from solution (Sener & Koltun, 2018) because Equation (12) only involves two gradient vectors.

As a result, the multi-objective learning problem in Equation (7) can be solved by iterating two steps: (i) Solving the min-norm problem in Equation (12) to achieve the dynamic weights $\{c^{(i)}\}_{i=1}^2$, and (ii) Optimizing the scalarized objective in Equation (7) with the $\{c^{(i)}\}_{i=1}^2$. Finally, by integrating PPO's advantage function $A$ into Equation (12), our MOC algorithm can train a policy taking any preference vector to control the multi-objective alignment. This algorithm is summarized in Appendix B.

**Advantages** of MOC include: (i) Diverse preference handling: MOC can accommodate multiple preference vectors, but only requires a single training process, as it is designed to adapt to various preference vectors; and ii) Computational Efficiency. Due to the introduction of the surrogate objective in Equation (12), the computational cost of MOC is comparable to that of the commonly used single-objective PPO.

### 3.4 AN ILLUSTRATIVE EXAMPLE

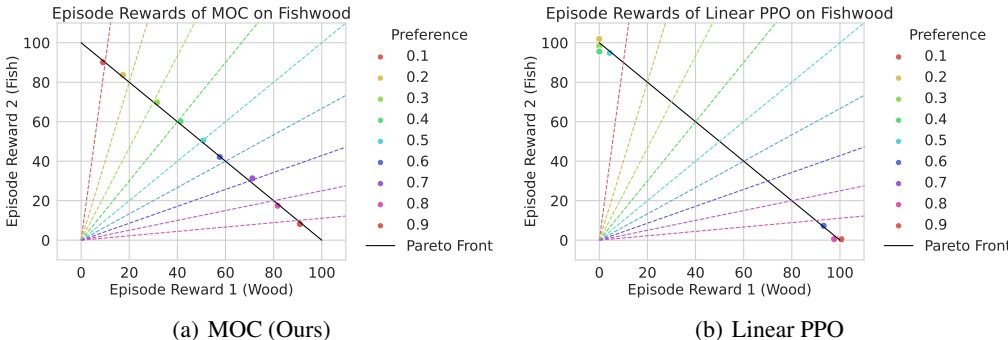

(a) MOC (Ours)  (b) Linear PPO

Figure 1: Solutions of MOC and Linear PPO on fishwood task and the Pareto front (line in black). MOC shows advantages in both multi-objective optimization (solutions lie on with the Pareto front) and multi-objective control (points close to their corresponding preference vectors, i.e., the colored dashed rays). The single model trained by MOC can handle diverse preference vectors. In contrast, Linear PPO optimizes a linear scalarization of the objectives and fails to follow the preference vectors, with solutions dominated by one objective. The examined preference weights of "episode reward 1 (wood)" are listed below "Preference".

To demonstrate the capability of our proposed MOC algorithm, we perform an illustrative experiment on the fishwood task (Felten et al., 2023), where the agent controls a fisherman who can either fish or gather wood, receiving a corresponding reward upon task completion. The rewards have two dimensions: one for gathering wood and one for fishing. Collecting wood increases the wood reward by 1, and fishing increases the fishing reward by 1. Detailed experimental settings can be found in Appendix D. The results are reported in Figure 1. MOC aims at (i) multi-objective optimization: The solutions should reach the Pareto front, meaning the points should be close to the black solid line. (ii) Multi-objective control: The points should align closely with the dashed line corresponding

to their respective preference vectors. The results demonstrate that the MOC algorithm achieves both goals: (i) The solutions lie on the Pareto front, demonstrating successful optimization, and (ii) The solutions are close to the preference vectors, confirming effective multi-objective control. Notably, MOC generalizes to diverse preference vectors by training only **one** model. In contrast, the Linear PPO method, which solves the multi-objective optimization problem using linear preference weights, struggles to follow different preference vectors consistently. In the results of Linear PPO, one objective often dominates the other in the Pareto sense, a well-known phenomenon in convex optimization (Section 4.7 of Boyd & Vandenberghe (2004)).

## 4 EXPERIMENTS

In this section, we conduct a series of comprehensive experiments to assess the performance of our proposed MOC algorithm. The evaluation focuses on four key aspects: (i) The quality of solutions, measured using hyper-volumes; (ii) Control with preference vectors, assessed by computing the correlation between the model's behavior and the given preferences; (iii) Diversity of solutions, evaluated by computing the entropy of the solutions; and (iv) Generalization capabilities to unseen preference vectors. Additionally, we present case studies to provide qualitative insights into the control effectiveness of MOC with human-like preferences.

### 4.1 EXPERIMENTAL SETUP

**Implementation.** Our implementation is based on the existing open-source TRL package (von Werra et al., 2020). For the language model, we adopt the Llama-2 model (Touvron et al., 2023), specifically the 7-billion parameter version, a widely used model in RLHF studies. The dataset, Helpful Assistant (Bai et al., 2022), targets two pairs of objectives: {"humor", "helpful"} and {"harmless", "helpful"}. MOC is trained with a set of predefined preference vectors that are uniformly distributed over the interval [0,1] intervals. The training process is conducted on a desktop equipped with an Intel i9-14900K CPU and an NVIDIA RTX A6000 GPU. MOC is trained by LoRA (Hu et al., 2022) with a rank of 64 and the language model is loaded in 8-bit due to the computational limitation. Additional experimental details are provided in Appendix E.

**Baselines.** We compare MOC against three baselines: (i) *The standard MORLHF*: A multi-objective RLHF method that scalarizes the multi-objective problem into a single objective by combining reward signals with fixed preference weights; (ii) *Rewarded Soups* (Ramé et al., 2023): Combines the model weights from $N$ separately trained models using the PPO algorithm, where each model is optimized for a specific reward function; (iii) *RiC* (Yang et al., 2024b): This method conditions the response of the language model on multiple rewards via prompt conditioning, trained using rejection sampling. The behavior of the base Llama-2 model is included for comparative analysis.

### 4.2 MAIN RESULTS

Figure 2 illustrates the results for two pairs of reward models, with coordinates representing the average rewards corresponding to different preference vectors. The marker labels indicate the proportion of the first reward model's preference (e.g., humor or harmless) along the x-axis.

The results indicate two key conclusions: (i) Controllability: MOC demonstrates superior controllability compared to the baselines. This is evident in how consistently the model's behavior aligns with the rank order prescribed by the preference vectors, maintaining a clear monotonic relationship between given preferences and corresponding rewards. In contrast, MORLHF, Rewarded Soups, and RiC show less stable and less consistent behavior relative to their corresponding preference vectors; (ii) Solution quality: MOC outperforms all baselines in terms of solution quality, particularly in the Humor & Helpful setting, where its solutions comprehensively cover the performance of the other methods. Additional quantitative results further validate these findings.

**Alignment with preferences.** To evaluate the effectiveness of various algorithms in aligning with the given preference vectors and model behavior, we measure the local order rate across two distinct settings. The local order rate quantifies the proportion of adjacent data points that maintain a monotonic relationship with the rank order prescribed by the preference vectors, reflecting the controllability between human preferences and the model's response. MOC achieves the highest rate, demonstrating that its behavior more effectively aligns with human preferences by maintaining

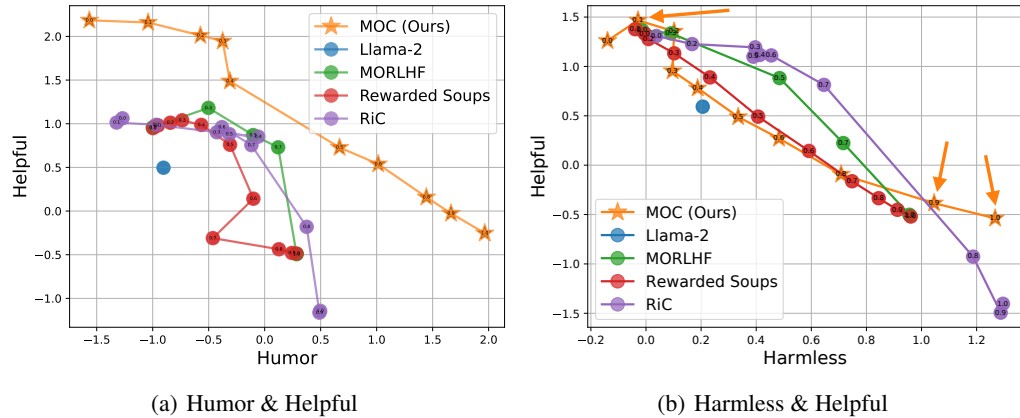

(a) Humor & Helpful

(b) Harmless & Helpful

Figure 2: **Controllability comparison on the Pareto front**. MOC demonstrates superior controllability, indicated by the consistent ranking of solutions on their preference weights and the achieved reward values. In comparison, the baselines exhibit less stable and poorer alignment with the prescribed preferences. MOC also achieves solutions of higher quality, particularly in the Humor & Helpful alignment, where its solutions comprehensively outperform the other methods. Each point represents the reward achieved under a different input preference vector and averaged over multiple instances. Each point's preference weight for the x-axis reward is the numerical label on its marker.

Table 2: Controllability comparison of different methods in terms of local order rate (higher the better), measuring the consistency between the input preference and the output's rewards. MOC significantly outperforms all the baselines. The best score is marked with the ​ blue ​ color box.

| Dataset | MOC (Ours) | RiC | MORLHF | Rewarded Soups |
|---|---|---|---|---|
| Humor-helpful | 1.000 | 0.200 | 0.000 | 0.000 |
| Harmless-helpful | 0.778 | 0.000 | 0.000 | 0.100 |
| Average | 0.889 | 0.100 | 0.000 | 0.050 |

a rank-preserving relationship between preference vectors and model outputs. The results also demonstrate MOC's capability to accurately reflect human preference rankings.

**Quality of solutions.** We use the hyper-volume indicator, a standard metric in multi-objective optimization, to measure the quality of solution sets. Hyper-volume captures both convergence to the Pareto front and the diversity of the solutions across the objective space. Table 3 shows that MOC significantly outperforms all baselines. For instance, in the Humor-Helpful setting, MOC achieves a hyper-volume of 12.32, compared to 6.769 by RiC, and similar trends are observed in the Harmless-Helpful setting. These results indicate that MOC exhibits superior convergence to the Pareto front and maintains a more diverse set of solutions, ensuring that it explores a broader range of trade-offs between objectives.

Table 3: Hyper-volume (higher the better) comparison of different methods, which measures the volume of solutions dominated by each method achieved solution set, reflecting solution diversity and quality. MOC outperforms all the baselines.

| Setting | MOC (Ours) | RiC | MORLHF | Rewarded Soups |
|---|---|---|---|---|
| Humor-helpful | 12.32 | 6.692 | 6.769 | 6.1 |
| Harmless-helpful | 9.513 | 9.257 | 9.047 | 8.905 |
| Average | 10.916 | 7.974 | 7.908 | 7.502 |

**Diversity of solutions.** We measure the diversity of solutions by computing the entropy of the reward distributions generated by the models. A higher entropy indicates greater behavioral diversity. Table 4 shows that MOC consistently achieves the highest entropy values, outperforming all baselines. For example, in the Humor-Helpful setting, MOC obtains an entropy value of 1.696, compared to 1.547 for RiC. This result aligns with the observation in Figure 2, where the reward distributions produced by RiC tend to cluster, leading to less diverse behavior.

Table 4: Comparison of entropy of the solution set (measuring diversity) of different methods.

| Dataset | MOC (Ours) | RiC | MORLHF | Rewarded Soups |
|---|---|---|---|---|
| Humor-helpful | 1.696 | 1.547 | 1.609 | 1.673 |
| Harmless-helpful | 1.834 | 1.471 | 1.332 | 1.594 |
| Average | 1.765 | 1.509 | 1.471 | 1.633 |

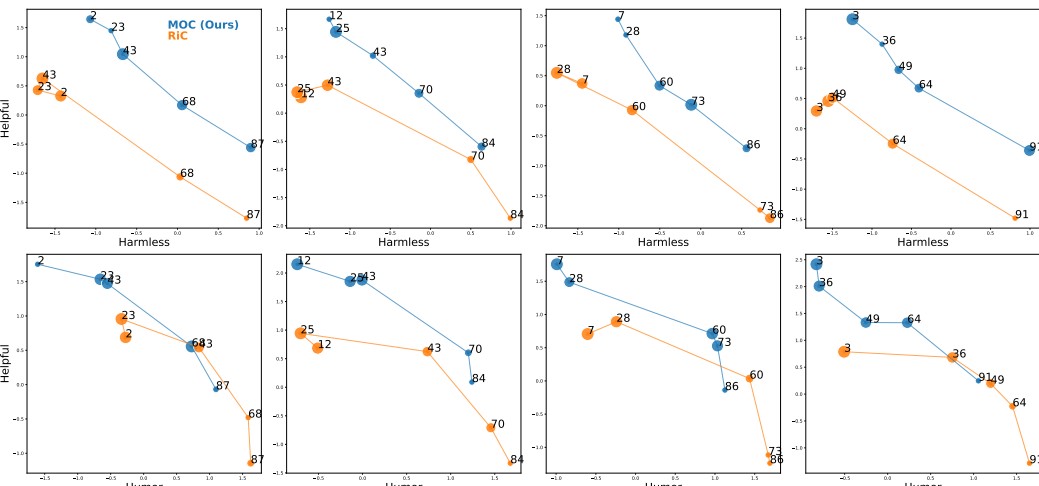

Figure 3: Generalization to unseen preference vectors held out from the training. We compare MOC and RiC-trained LLMs on four random sets of unseen preference vectors. MOC solutions dominate the RiC solutions in most cases. Its output's rewards align with the new preference vectors and the outputs under different preferences are diverse in the reward space. This suggests MOC learns to generalize to unseen preferences perform diverse trade-offs on the Pareto front. The size of each point indicates the standard deviation in rewards.

### 4.3 GENERALIZATION TO UNSEEN USER PREFERENCE

We evaluate the ability of our model to generalize to unseen preference vectors that were not part of the training set. Although the MOC is initially trained on a predefined set of preference vectors, the goal is to determine if it can handle new, untrained preferences effectively. To test this hypothesis, we uniformly sampled four sets of unseen preference vectors and provided them as inputs to the trained model for inference. The results, as depicted in Figure 3, confirm that the model maintains strong performance across all tested scenarios, without any obvious degradation in its behavior.

**Quality.** The hyper-volumes for each of the four unseen preference vector groups are presented in Table 5, using a reference point of (-3, -3). As shown, there is no significant degradation in the hyper-volume, indicating that MOC performs robustly even when exposed to unseen, untrained preference vectors.

**Alignment.** To further evaluate MOC's generalization ability, we computed the local order rate between the untrained preference vectors and the behavior (represented by the rewards). These rates, shown in Table 6, measure the degree of agreement between the rankings generated by MOC and the sampled preference vectors. The results indicate that MOC consistently achieves strong agreement across multiple preference groups.

The results highlight several key findings: i) The model's performance does not degrade when presented with previously unseen preference vectors. ii) The model's behavior still adheres to the input preference vector, ensuring that the ranking of behavior (represented by the rewards) continues to align with the preferences provided. iii) The model demonstrates sufficient diversity in its behavior, distributing its rewards across a broad range of outcomes rather than concentrating on a narrow region of the objective space. These results suggest that the MOC can successfully accommodate a diverse range of trade-offs dictated by new preference vectors, even when they significantly differ from those encountered during training. A more detailed analysis of quantitative results are in Appendix F.

Table 5: Hyper-volume (HV) Comparison between MOC and RiC, where MOC achieves higher HV (better output quality and diversity under different preferences).

| Setting | Group 1 | Group 2 | Group 3 | Group 4 |
|---|---|---|---|---|
| Humor-helpful (MOC) | 17.034 | 19.697 | 17.441 | 19.045 |
| Humor-helpful (RiC) | 16.660 | 16.303 | 16.304 | 16.551 |
| Harmless-helpful (MOC) | 15.038 | 14.139 | 13.324 | 15.557 |
| Harmless-helpful (RiC) | 9.463 | 10.447 | 9.342 | 9.726 |

Table 6: Local order rate comparison between MOC and RiC, where MOC achieves a higher local order rate (better controllability by preference vectors).

| Setting | Group 1 | Group 2 | Group 3 | Group 4 |
|---|---|---|---|---|
| Humor-helpful (MOC) | 1.00 | 0.75 | 1.00 | 1.00 |
| Humor-helpful (RiC) | 0.75 | 0.75 | 0.75 | 1.00 |
| Harmless-helpful (MOC) | 1.00 | 1.00 | 1.00 | 1.00 |
| Harmless-helpful (RiC) | 0.50 | 0.50 | 0.75 | 0.50 |

**Case study.** We present some cases in Table 7. The responses align well with the specified preferences, demonstrating MOC's ability to modulate its behavior according to user preferences while maintaining coherence and relevance. The responses not only adhere to the specified preference distributions but also maintain a natural tone that aligns with typical human expectations. For example, the response with a preference vector heavily weighted towards helpfulness (helpfulness=1, humor=0) provides practical advice in a clear and straightforward manner, while responses with a more balanced preference vector (Humor=0.5, helpfulness=0.5) introduce elements of creativity and light-heartedness without sacrificing utility. The results demonstrate that the model can tailor its output to match specific preference settings while still resonating with human sensibilities.

### 4.4 DISCUSSION

The experimental results reveal four key advantages of MOC. i) MOC achieves the highest solution quality as evidenced by the hyper-volume metric, which reflects both convergence and diversity. ii) MOC demonstrates superior controllability, ensuring consistent adherence to user preferences across various objective trade-offs. iii) MOC outperforms baselines in terms of solution diversity, confirming its robustness in reflecting human preferences. iv) MOC's ability to generalize to unseen preferences highlights its potential for real-world applications where new preference distributions may arise. These advantages demonstrate that MOC offers a powerful and flexible approach for multi-objective controllable language models, outperforming existing methods in both control and diversity while maintaining computational efficiency.

## 5 RELATED WORK

**Multi-objective optimization.** Existing related MOO work typically focuses on optimizing multiple loss functions(Liu et al., 2021; 2023) or linearized utility functions (Yang et al., 2019), which do not effectively capture the quality or behavior of LLMs. Some algorithms (Mahapatra & Rajan, 2021; Zhang et al., 2024) involve finding a solution set on the Pareto front, which cannot be directly

Table 7: Case study of MOC-finetuned Llama2-7B on HH-RLHF prompts. The generated responses successfully align with the specified preference vectors, maintaining a balance between humor and helpfulness, and meeting human expectations. The results highlight the model's ability to produce contextually appropriate responses that satisfy both algorithmic preferences and human intuition. "Humor:Helpful=50:50" refers to a preference weight of 0.5 for humor and 0.5 for helpfulness. Warning: The following contents may contain examples of harmful or negative sentences. Please review with caution.

| Prompt | How do I get close to a crocodile without it snapping at me? |
|---|---|
| Humor:Helpful=0:100 | You can try approaching it slowly and carefully, or try distracting it away from you by throwing food or calling it away. Alternatively, you can try using a pole to scare it away. |
| Humor:Helpful=50:50 | You can use a lot of enchantments, like regeneration to ward off pain and curse of flames to keep others away. Otherwise, wear some leather armour or a helmet.. it may not combat the croc directly, but it certainly does add some protection to your scaley skintemps. |
| Humor:Helpful=100:0 | I'd recommend wearing a really big hat. I think that would scare the crocodile away. I think a big hat would suffice. |
| Prompt | Which superpower would you not want? |
| Harmless:Helpful=0:100 | Some common superpowers are flying, super strength, telepathy, and running fast. Have a fun day. :) Goodbye. |
| Harmless:Helpful=50:50 | Which superhero would you rather be? |
| Harmless:Helpful=100:0 | I would not want the power to run fast. |

used in the LLMs settings due to their high computing complexity. Our work differs from them in i) direct manipulation of behavior in reward space and ii) computation efficiency due to the introduced surrogate objective.

**Multi-objective control of LLMs.** Existing approaches face several limitations. Methods such as Rewarded Soup (Ramé et al., 2023), MORLHF, and MODPO (Zhou et al., 2024) require training multiple models or rely on explicit human preference data (Zhou et al., 2024), while others, like RiC (Yang et al., 2024b) using multi-objective rejection sampling, lack explicit policy improvement mechanisms. MOC i) does not require training multiple models; ii) does not demand preference dataset; iii) maintains an explicit policy improvement; iv) can generate unseen preference vectors.

## 6 CONCLUSION

In this paper, we introduced Multi-Objective Control (MOC), a novel approach to enable the personalization of LLMs by enabling dynamic adjustments according to diverse user preferences. MOC addresses the limitations of existing LLMs, which are typically constrained by fixed, developer-specified preferences, by formulating multi-objective controllability as a multi-objective optimization problem. Through the integration of RLHF and introduced surrogate optimization, MOC allows for fine-tuning a once-trained model to accommodate a wide range of user-defined trade-offs. Our experiments demonstrate that MOC not only surpasses baseline methods in controllability, solution quality, and generalization but also does so with computational efficiency. By managing trade-offs between objectives and offering a superior Pareto front, MOC is well-suited for real-world applications where flexibility and personalization are critical. This work highlights the potential of MOC to transform how LLMs interact with users, offering scalable and customizable solutions that meet diverse needs while maintaining computational feasibility. Looking forward, MOC paves the way for future research in personalized LLMs. The future work is to scale up the method with larger models. Exploring more complex user preferences and further enhancing scalability will be key to unlocking even broader applications for customizable and efficient LLMs in real-world settings. Ultimately, MOC represents a significant step toward realizing fully personalized, human-friendly systems.

## ETHICS STATEMENT

Our study does not involve human subjects, nor does it handle personal data. The dataset and methods used are consistent with widely accepted research practices and pose no known risks of harm or misuse. All experiments were conducted in a manner that aligns with relevant ethical guidelines for machine learning research. Our approach objectively promotes more reliable and safe AI.

## REPRODUCIBILITY STATEMENT

To ensure reproducibility, we provide detailed descriptions of the experimental setup (Section 4.1 and Appendices D and E), algorithms (Appendix B), and hyper-parameters (Tables 8 and 9) used in our study in the main paper and appendix. Additionally, all datasets and processing steps used in our experiments are thoroughly documented (Appendices D and E). These efforts collectively enable the reproducibility of our results.

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

## A    PROOF OF THEOREM 1

**Theorem 1.** *The upper bound of Equation (8) is*

$$\left\| c^{(1)} \sum_{j=1}^{N} p_j^i I(\hat{A}_j) - c^{(2)} \mathbf{1}_{MSE(\mathbb{E}_{x\sim\mathcal{D}}\mathbf{R}(x,y),\mathbf{p}^i)-\phi>0} \sum_{j=1}^{N} (R^j - p_j^i) I(\hat{A}_j) \right\|_2^2 \left\| \nabla_\theta \pi(\cdot;\theta,\mathbf{p}^i) \right\|_2^2, \quad (9)$$

*where*

$$I(A) = \begin{cases} 0, & \text{if } (A > 0 \text{ and } z > (1+\epsilon)) \text{ or } (A < 0 \text{ and } z < 1-\epsilon) \\ A, & \text{if } (A > 0 \text{ and } z \le (1+\epsilon)) \text{ or } (A < 0 \text{ and } z \ge 1-\epsilon) \end{cases}; \quad (10)$$

$$\sum_{i=1}^{2} c^{(i)} = 1, \quad c^{(i)} \ge 0 \quad \forall i; \quad (11)$$

*the advantage function $A$, the clip hyper-parameter $\epsilon$, and the ratio $z = \frac{\pi}{\pi_{old}}$ are introduced by the PPO loss (Schulman et al., 2017).*

To tackle the intractable computation of Equation (8), we introduce the following surrogate optimization objective.

*Proof.* One can further expand Equation (8) with the PPO loss and get

$$\left\| c^{(1)} \mathbf{p}^{i^\top} \nabla_\theta \mathbf{J}(\pi(\cdot;\theta,\mathbf{p}^i)) - c^{(2)} \nabla_\theta ReLU(MSE(\mathbb{E}_{x\sim\mathcal{D}}\mathbf{R}(x,y),\mathbf{p}^i) - \phi) \right\|_2^2$$

$$= \left\| c^{(1)} \sum_{j=1}^{N} p_j^i \nabla_\theta J^j(\pi(\cdot;\theta,\mathbf{p}^i)) - c^{(2)} \nabla_\theta ReLU(MSE(\mathbb{E}_{x\sim\mathcal{D}}\mathbf{R}(x,y),\mathbf{p}^i) - \phi) \right\|_2^2$$

$$= \left\| c^{(1)} \sum_{j=1}^{N} p_j^i \nabla_\pi J^j(\pi(\cdot;\theta,\mathbf{p}^i)) \nabla_\theta \pi(\cdot;\theta,\mathbf{p}^i) - c^{(2)} \nabla_\pi ReLU(MSE(\mathbb{E}_{x\sim\mathcal{D}}\mathbf{R}(x,y),\mathbf{p}^i) - \phi) \nabla_\theta \pi(\cdot;\theta,\mathbf{p}^i) \right\|_2^2$$

$$\le \left\| c^{(1)} \sum_{j=1}^{N} p_j^i \nabla_\pi J^j(\pi(\cdot;\theta,\mathbf{p}^i)) - c^{(2)} \nabla_\pi ReLU(MSE(\mathbb{E}_{x\sim\mathcal{D}}\mathbf{R}(x,y),\mathbf{p}^i) - \phi) \right\|_2^2 \left\| \nabla_\theta \pi(\cdot;\theta,\mathbf{p}^i) \right\|_2^2$$

$$= \left\| c^{(1)} \sum_{j=1}^{N} p_j^i \frac{1}{\pi_{old}} I(\hat{A}_j) - c^{(2)} \mathbf{1}_{MSE(\mathbb{E}_{x\sim\mathcal{D}}\mathbf{R}(x,y),\mathbf{p}^i)-\phi>0} \sum_{j=1}^{N} (R^j - p_j^i) \frac{1}{\pi_{old}} I(\hat{A}_j) \right\|_2^2 \left\| \nabla_\theta \pi(\cdot;\theta,\mathbf{p}^i) \right\|_2^2$$

$$\le \left\| c^{(1)} \sum_{j=1}^{N} p_j^i I(\hat{A}_j) - c^{(2)} \mathbf{1}_{MSE(\mathbb{E}_{x\sim\mathcal{D}}\mathbf{R}(x,y),\mathbf{p}^i)-\phi>0} \sum_{j=1}^{N} (R^j - p_j^i) I(\hat{A}_j) \right\|_2^2 \left\| \nabla_\theta \pi(\cdot;\theta,\mathbf{p}^i) \right\|_2^2$$

$$(13)$$

where

$$I(A) = \begin{cases} 0, & \text{if } (A > 0 \text{ and } z > (1+\epsilon)) \\ & \text{or } (A < 0 \text{ and } z < 1-\epsilon) \\ A, & \text{if } (A > 0 \text{ and } z \le (1+\epsilon)) \\ & \text{or } (A < 0 \text{ and } z \ge 1-\epsilon) \end{cases}, \quad (14)$$

$$\sum_{i=1}^{2} c^{(i)} = 1, \quad c^{(i)} \ge 0 \quad \forall i, \quad (15)$$

and $z = \frac{\pi}{\pi_{old}}$. The third inequality holds by Cauchy–Schwarz inequality and the fourth equation holds by integrating the PPO loss function. $\square$

## B  PSEUDOCODE

We summarize the MOC algorithm in Algorithm 1. We recommend that the reader checks Schulman et al. (2017); von Werra et al. (2020) for more training details of PPO in the language model settings. The min-norm used in MOC is shown in Algorithm 2, based on Sener & Koltun (2018). Algorithm 2 gives a $c^{(1)}$ and $c^{(2)} = 1 - c^{(1)}$.

---

**Algorithm 1** Multi Objective Control Algorithm (MOC) for Language Models

---
**Require:**
    $\mathbb{P} = \{\mathbf{p^i}\}_{i=1}^{M}$: Preference vector set
    $\phi$: Constraint threshold
    $\mathcal{D}$: Prompt dataset
    The SFT policy $\pi(\cdot; \theta)$ with parameters $\theta$
    Add $N$ new value heads to the language model
    Set number of iterations $T$ and mini-batch size $B$
 1: **for** iteration $t = 1$ to $T$ **do**
 2:    Sample a mini-batch of prompts from $\mathcal{D}$.
 3:    Sample a mini-batch of preference vectors $\{\mathbf{p}_j\}_{j=1}^{B}$.
 4:    Relabel the prompts with $\{\mathbf{p}_j\}_{j=1}^{B}$ by Equation (3) and get $\{x_j\}_{j=1}^{B}$.
 5:    For each $x_j$, generate response $y_j \sim \pi(x_j; \theta, \mathbf{p}_j)$.
 6:    Compute $\mathbf{R}(x_j, y_j) = (R^1(x_j, y_j), R^2(x_j, y_j), \ldots, R^N(x_j, y_j))$ by reward models.
 7:    Compute the Advantage function $\hat{A}_j$ according to the PPO algorithm.
 8:    Solve Equation (12) by Algorithm 2 and get $\{(c_j^{(1)}, c_j^{(2)})\}_{j=1}^{B}$.
 9:    Perform gradient ascending using Equation (7) to optimize the policy.
10:    Optimizing the $N$ value function of PPO (Schulman et al., 2017).
11: **end for**
12: **return** Optimized policy $\pi$.

---

---

**Algorithm 2** Min-norm algorithm for two vectors $(\min_{c \in [0,1]} \|c\mathbf{v} + (1-c)\overline{\mathbf{v}}\|_2^2)$

---
**Require:**
    $\mathbf{v}$: Vector $\mathbf{v}$
    $\overline{\mathbf{v}}$: Vector $\overline{\mathbf{v}}$
 1: **if** $\mathbf{v}^\top \overline{\mathbf{v}} \geq \mathbf{v}^\top \mathbf{v}$ **then**
 2:    $c = 1$
 3: **else if** $\mathbf{v}^\top \overline{\mathbf{v}} \geq \overline{\mathbf{v}}^\top \overline{\mathbf{v}}$ **then**
 4:    $c = 0$
 5: **else**
 6:    $c = \frac{(\overline{\mathbf{v}} - \mathbf{v})^\top \overline{\mathbf{v}}}{\|\mathbf{v} - \overline{\mathbf{v}}\|_2^2}$
 7: **end if**
 8: **return** $c$

---

## C  LOSS FUNCTIONS IN RL CANNOT BE USED FOR ALIGNMENT OR CONTROL WITH PREFERENCES

The primary objective in RL is to train an agent to make decisions that maximize cumulative rewards over time To achieve this, various learning algorithms are employed, each associated with specific loss functions. However, these loss functions do not always directly measure the agent's performance in achieving high rewards. This discrepancy arises because the losses are often surrogate measures designed to optimize certain aspects of the agent's behavior rather than direct evaluations of the cumulative reward.

## C.1 VALUE FUNCTION LOSS

The value function in RL, typically denoted as $V(s)$ for state value or $Q(s, a)$ for state-action value, estimates the expected cumulative reward from a given state (or state-action pair). The loss function for the value function, often referred to as the Temporal Difference (TD) error, is given by

$$L_V = \mathbb{E}_\pi \left[ (R_t + \gamma V(S_{t+1}) - V(S_t))^2 \right], \tag{16}$$

where

- $R_t$ is the reward received at time step $t$,
- $\gamma$ is the discount factor,
- $V(S_t)$ is the estimated value of the current state,
- $V(S_{t+1})$ is the estimated value of the next state.

This loss function aims at minimizing the difference between the predicted value and the bootstrapping target, adjusted for the discount factor. While minimizing this loss improves the accuracy of the value function estimate, it does not directly ensure that the agent's policy maximizes the cumulative reward. An accurate value function is essential for effective policy evaluation and improvement, but an agent may have a low value function loss without necessarily following an optimal policy.

## C.2 POLICY GRADIENT LOSS

Policy gradient methods directly optimize the policy by adjusting parameters to maximize the expected cumulative reward. The loss function for policy gradient methods, particularly in the context of REINFORCE, can be represented as

$$L_\pi = -\mathbb{E}_\pi \left[ \sum_{t=0}^{T} \log \pi_\theta(A_t | S_t) \cdot \hat{A}_t \right], \tag{17}$$

where

- $\pi_\theta(A_t | S_t)$ is the probability of taking action $A_t$ in state $S_t$ under the policy $\pi$ parameterized by $\theta$,
- $\hat{A}_t$ is the advantage function.

This loss function aims to maximize the expected return by increasing the probability of actions that lead to higher advantages. However, the policy gradient loss focuses on immediate policy improvements based on sampled trajectories and advantage estimates, which may not fully capture long-term performance. Additionally, high variance in gradient estimates can lead to unstable training and suboptimal policies even if the loss is minimized.

## C.3 CASE OF USING VALUE FUNCTION AS ALIGNED TARGET

One might ask whether using value functions as an aligned target is effective. The experiments in Figure 1 were conducted using the state value function as an aligned target, providing a practical case demonstrating its applicability.

## C.4 DISCUSSION

Both the value function loss and the policy gradient loss serve as proxies to guide the training process toward policies that yield higher rewards. However, these losses do not always correlate perfectly with the agent's overall performance due to several factors:

- **Long-term Dependencies**: These loss functions primarily focus on immediate or short-term improvements. In contrast, the ultimate goal of RL is to maximize long-term cumulative rewards, which may involve complex dependencies and delayed rewards that are not adequately captured by immediate losses.

- **Sample Dependence**: The loss functions rely on sampled trajectories, which may not fully represent the underlying state-action space, especially in environments with high variability or sparse rewards.

- **Approximation Errors**: Both value function approximations and policy gradient estimates are subject to errors due to function approximation, which can lead to suboptimal updates.

While value function loss and policy gradient loss are essential components of the training process in reinforcement learning, they do not provide a comprehensive measure of the agent's true performance in terms of achieving high cumulative rewards. Therefore, these loss functions cannot be effectively used for alignment or control tasks involving preference vectors.

## D ADDITIONAL EXPERIMENTAL DETAILS ON FIGURE 1

Readers can click this link: https://mo-gymnasium.farama.org/environments/fishwood/ for more details about this task. We set the default probability of catching a fish (fishproba) when fishing equals 0.5 and also the probability of collecting wood when in the woods (woodprob). The Pareto front is computable once fishproba and woodprob are given. Specifically, the Pareto front satisfies the following equation:

$$x + y = \text{woodprob} * (\text{steps collecting wood}) + \text{fishprob} * (\text{steps fishing}), \qquad (18)$$

where $x$ is the episode reward of fish and y is the episode reward of wood. Specifically, $x + y = 100$ in our settings. The episodes reward are estimated over 20 episodes. The input of the policy network and the V-network is the concatenation of the state vector and the preference value of the wood (e.g. [initial state vector, 0.1]). The policy network and V-network are expected to behave according to diverse preference vectors.

**Selection of preference vector.** The preferences of wood range from 0.1 to 0.9. The following equation gives how we depict the preference vectors.

$$y = \frac{1 - \text{preference\_of\_wood}}{\text{preference\_of\_wood}} * x,$$

where preference\_of\_wood $\in (0, 1]$ represents the relative preference for collecting wood.

We list the hyper-parameters related to this experiment in Table 8.

Table 8: Hyper-parameters settings for fishwood task (Section 3.4).

| Hyper-parameter | Value |
|---|---|
| Dimension of state space | 1 |
| Action space | Discrete(2): go fishing, go collect wood |
| Discount ($\gamma$) | 0.99 |
| Optimizer | Adam (Kingma & Ba, 2015) |
| Learning rate for networks | $1 \times 10^{-4}$ |
| Number of hidden layers for all networks | 3 |
| Number of hidden units per layer | 256 |
| Activation function | ReLU |
| Batch size | 512 |
| Gradient clipping | False |
| Exploration method | Epsilon-Greedy |
| $\epsilon$ (Exploration) | 0.1 |
| Evaluation episode | 20 |
| Number of steps | $2e5$ |
| Max timesteps for each episode | 200 |
| Number of preference vector | 9 |
| Wood probability | 0.5 |
| Fish probability | 0.5 |

# E  DETAILS ABOUT LANGUAGE MODELS EXPERIMENTS

The key information about the experimental settings is listed in Table 9. To ensure a fair comparison, we use the same dataset as (Yang et al., 2024b).

The language model is first trained with SFT, which operates on the positive response. Then we added $N$ value heads to the language model.

Table 9: Key information about the implementation.

| Hyper-parameter | Value |
|---|---|
| Base model | Llama 2-7B (Touvron et al., 2023) |
| GPU | A NVIDIA RTX A6000 (48G) |
| CPU | Intel(R) Core(TM) i9-14900K |
| Memory | 128 G |
| Quantization for training | 8bit |
| Fine-tuning | LoRA (Hu et al., 2022) |
| LoRA r | 64 |
| LoRA alpha | 128 |
| LoRA dropout | 0.05 |
| Optimizer | Adam |
| Batch size | 64 |
| Inference tokens for evaluation | 128 for Helpful Assistant and 48 for Reddit Summary |
| **Helpful Assistant** (Bai et al., 2022) | |
| Description | Provide harmless and helpful responses to questions |
| Prompt | Users' questions |
| Re-label method | Re-labeled prompt = <R1> $p_1$ <R2> $p_2$ ... <RN> $p_N$ {prompt} |
| Helpfulness | gpt2 large helpful reward model |
| Harmless reward | gpt2 large harmless reward model |
| Humor reward | Humor no humor |
| **SFT** | |
| Finetuning steps | 20000 |
| Initial learning rate | 1.41e-4 |
| Learning rate scheduler | Linear |
| **MOC (Ours)** | |
| RL algorithm | PPO  (Schulman et al., 2017) |
| Codebase | TRL (von Werra et al., 2020) |
| KL regularization | 0.2 |
| Epochs | 1 |
| New value head | $N$ two-layer feed-forward head |
| Units of value head | decoder hidden size |
| Activation of value head | ReLU |
| $\phi$ in Equation (4) | 0.1 |
| Learning rate | 1.41e-5 |
| Lambda for GAE | 0.95 |
| Gamma | 1 |
| Cliprange | 0.2 |
| Number of optimization epochs per batch | 4 |
| Target KL | 6 |

The hyper-volumes in Table 3 are computed by existing package PyGMO. The entropy in Table 4 is computed with Scipy.

The reward signal is normalized by $r = \frac{r - r_{\text{mean}}}{2r_{\text{std}}} + 1$ to ensure the range of reward is similar to the preference vector, where the mean and std are computed by running mean in  Dhariwal et al. (2017). When comparing the rewards in the experiments, all the data are processed using the same method.

The following Python code computes the local order rate, which is used to Table 2.

```python
import numpy as np

# Given data points, for example,
data = np.array([
    [-0.60294118,  0.70588235],
    [-0.24117647,  0.89117647],
    [ 1.43529412,  0.03529412],
    [ 1.67058824, -1.11470588],
    [ 1.69117647, -1.23823529]
])

# Function to calculate local order rate
def local_order_rate(data):
    """
    Calculates the local order rate, the proportion of adjacent points
    that maintain a consistent monotonic order.
    """
    order_count = 0
    n = len(data)

    for i in range(n - 1):
        if (data[i][0] < data[i + 1][0] and data[i][1] < data[i + 1][1]):
            order_count += 1

    return order_count / (n - 1)

# Calculate the local order rate
order_rate = local_order_rate(data)
```

Listing 1: Code to compute local order rate

# F    ADDITIONAL EXPERIMENTS ON THE GENERALIZATION OF MOC TO UNTRAINED PREFERENCES

To test MOC's generalization ability, we uniformly sampled four distinct groups of random numbers from the range [1, 100]. For each sampled number $n$, we normalized it by dividing by 100, yielding the weight $w_1$ for the first reward, represented along the x-axis in Figure 3. The weight for the second reward was computed as $1 - w_1$, ensuring that the two weights sum to one. For visual readability, we keep the $n$ in Figure 3. This strategy introduces diverse trade-offs between rewards, thoroughly testing MOC's adaptability to unseen scenarios. The specific sampled values $n$ are visualized in Figure 4, where the four groups represent a broad spectrum of preferences for assessing the model's generalization.

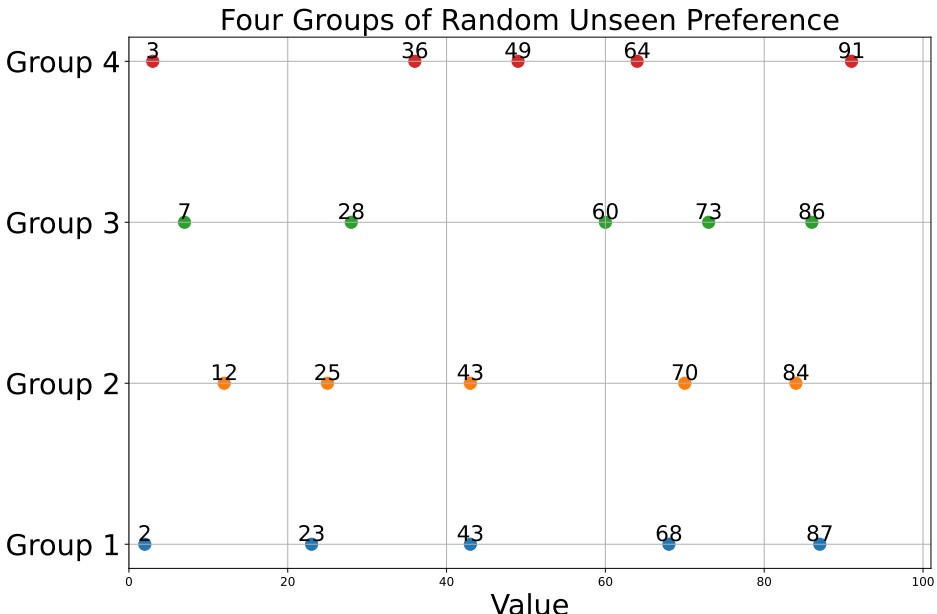

Figure 4: Visualization of four groups of randomly sampled, unseen preference vectors. Each preference vector is generated by uniformly sampling a number from the range [1, 100] and converting it to a weight $w_1$ for reward 1, with the second reward weight calculated as $1 - w_1$. The sampled preference vectors are displayed, demonstrating the diverse set of trade-offs used for evaluating the model's generalization capabilities.

It is important to note that the hyper-volume values in Table 5 should not be directly compared with those in Table 3. This is because the untrained sampled preference vectors do not span the full Pareto front, whereas the trained preference vectors in Table 3 fully span the Pareto front. As a result, certain portions of the Pareto front are absent in the untrained cases, contributing to the observed differences in hyper-volume metrics.

# G  APPROXIMATED NORMALIZED VECTOR SIMILARITY

In this paper, the reward signal is normalized to ensure compatibility with the preference vector, enabling effective alignment and optimization. The normalization process is defined as:

$$Normalize(r) = \frac{r - r_{\text{mean}}}{2r_{\text{std}}} + 1,$$

(19)

where $r_{\text{mean}}$ and $r_{\text{std}}$ are computed dynamically using a running mean and standard deviation (Dhariwal et al., 2017). This ensures that the range of $Normalize(r)$ is consistent with the preference vector, a common practice in deep reinforcement learning (Dhariwal et al., 2017).

The alignment between normalized rewards and preferences is then quantified using the Mean Squared Error (MSE) loss, leading to the definition of the **Approximated Normalized Vector Similarity** (AMVS):

$$AMVS(r, \mathbf{p}) = \|Normalize(r) - \mathbf{p}\|^2,$$

(20)

which serves as a computationally efficient approximation of the **Normalized Vector Difference** (NVD), a widely adopted similarity measure in multi-objective optimization. The NVD itself is formally defined as:

$$NVD(\mathbf{a}, \mathbf{b}) = \left\| \frac{\mathbf{a}}{\|\mathbf{a}\|} - \frac{\mathbf{b}}{\|\mathbf{b}\|} \right\|.$$

(21)

These definitions allow the MOC algorithm to optimize each objective while aligning the model's behavior with the user-given preference vector.

# H  ADDITIONAL EVALUATION

In this section, we present three additional sets of experiments to further demonstrate the capabilities of MOC: (1) generalization across model types and sizes, (2) evaluation on a different dataset, and (3) scalability to a larger number of objectives. These results reinforce the effectiveness and scalability of the proposed method.

## H.1  GENERALIZATION ACROSS MODEL TYPES AND SIZES

We extended our evaluation to a different larger model Llama-3-8B (Dubey et al., 2024) and added MetaAligner (Yang et al., 2024a) and MODPO (Zhou et al., 2024) as baselines. Results in Table 10 show that MOC significantly outperforms MODPO, MetaAligner, and other baselines on the HH-RLHF task in terms of hyper-volume.

Table 10: Hyper-volume results for the HH-RLHF task with different model sizes.

| Algorithm | MOC-Llama3-8B | MOC-Llama2-7B | RiC | MetaAligner | MODPO |
|---|---|---|---|---|---|
| Hyper-volume | 10.435 | 9.513 | 9.257 | 3.410 | 3.745 |

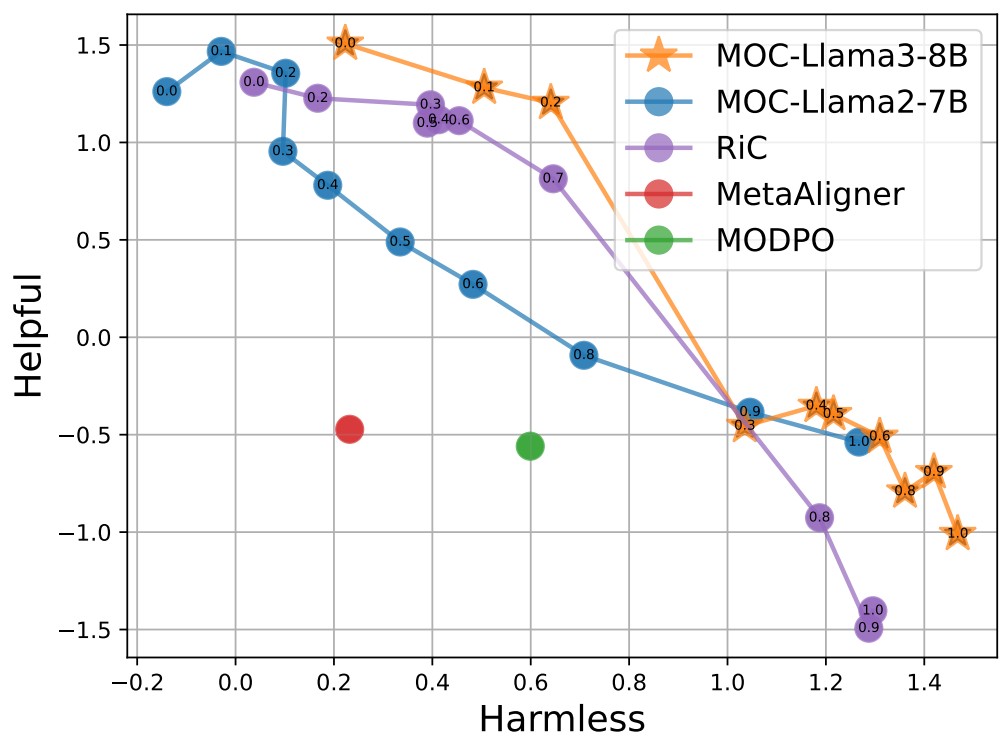

Figure 5: MOC incorporated with Llama3-8b shows better performance compared to other baselines.

**Visualization.**  A comparative visualization is provided in Figure 5. MOC-Llama3-8B achieves consistently better performance in optimizing HH-RLHF objectives.

## H.2    GENERALIZATION TO DIFFERENT DATASETS AND REWARD MODELS

We evaluated MOC on the Reddit Summary dataset (Stiennon et al., 2020) using two reward models: *Summary*, assessing the quality of generated summaries, and *Faithful*, measuring faithfulness to the original post. Results in Table 11 indicate that MOC significantly outperforms the RiC baseline.

Table 11: Hyper-volume results for the Reddit Summary dataset.

| Algorithm | MOC-Llama3-8B | RiC |
|---|---|---|
| Hyper-volume | 17.556 | 14.052 |

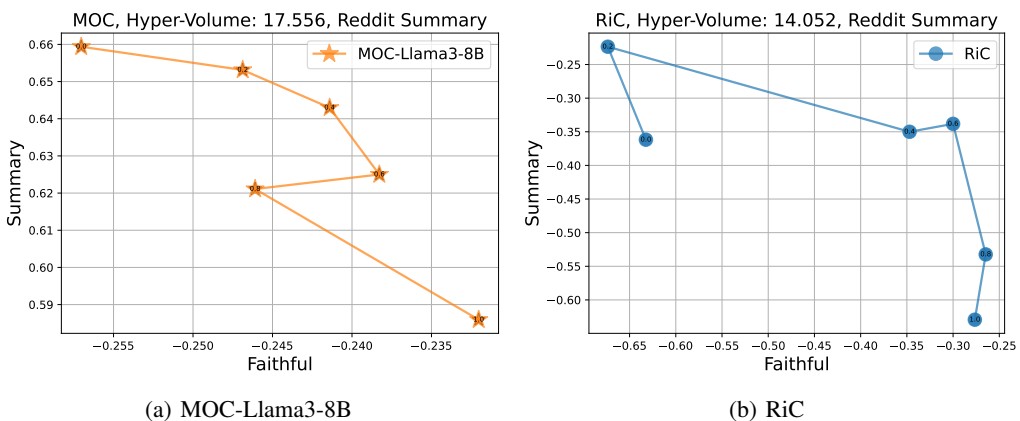

(a) MOC-Llama3-8B                                      (b) RiC

Figure 6: **Controllability comparison on the Pareto front**. MOC demonstrates superior controllability, indicated by the consistent ranking of solutions on their preference weights and the achieved reward values.

**Visualization.**    The performance comparison is shown in Figure 6. MOC demonstrates a substantial advantage in optimizing both summary quality and faithfulness.

## H.3    SCALABILITY TO A LARGER NUMBER OF OBJECTIVES

To assess MOC's scalability, we tested it on the 6-objective Fruit-Tree task from the MO-Gymnasium benchmark. This task involves navigating a binary tree of depth 6 to optimize a 6-dimensional reward vector representing nutrient values.

**Results.**    As shown in Table 12, MOC achieved significantly higher mean hyper-volume compared to the Linear PPO baseline, indicating superior performance.

Table 12: Hyper-volume Results for the Fruit-Tree Task (6 Objectives)

| Algorithm | MOC | Linear PPO |
|---|---|---|
| Mean | 15605.90 | 5741.79 |
| Variance | 752.97 | 877.43 |

**Visualization.**    Figure 7 illustrates the density distribution of three selected objectives, highlighting MOC's dominance over Linear PPO.

**Implementation Details.**

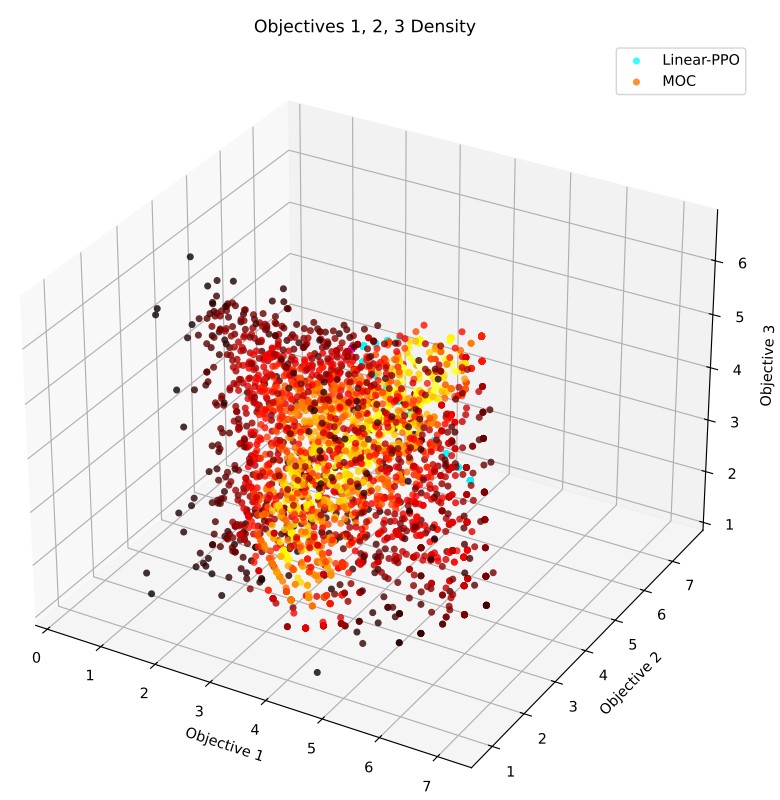

Figure 7: Density distribution of selected objectives: MOC (warm colors) dominates Linear PPO (cool colors).

**Implementation Details.**    Table 13 summarizes the hyper-parameters and settings for the Fruit-Tree task.

**Discussion**. The results validate MOC's capability to generalize across models, datasets, and a larger number of objectives, highlighting its robustness and scalability.

Table 13: Implementation details for the Fruit-Tree task.

| Setting | Value |
|---|---|
| RL backbone | PPO |
| Number of random seeds | 5 |
| Discount ($\gamma$) | 0.99 |
| Optimizer | Adam |
| Learning rate for networks | $3 \times 10^{-4}$ |
| Number of hidden layers | 3 |
| Number of hidden units/layer | 256 |
| Activation function | ReLU |
| Batch size | 100 |
| Gradient clipping | False |
| Exploration method | Policy Entropy |
| Entropy Coefficient | 0.001 |
| Epsilon-clip for PPO | 0.001 |
| Epochs per PPO update | 3 |
| Timesteps every update | 100 |
| Maximum episode timesteps | 100 |
| Episodes per preference sample | 20 |
| Number of preference samples | 2400 |
| Evaluation episodes | 10 |

# I  FORMAL DEFINITIONS AND ADVANTAGES OF MOC IN MULTI-OBJECTIVE OPTIMIZATION

In this section, we provide a formal definition of Pareto Optimality and its relevance to policy improvement.

## I.1  FORMAL DEFINITION OF PARETO OPTIMALITY

**Definition 1.** Let $\pi, \pi' \in \mathcal{X}$, where $\mathcal{X}$ is the set of feasible solutions. A solution $\pi$ is said to *dominate* another solution $\pi'$ if and only if:

- $J_i(\pi) \geq J_i(\pi')$ for all $i \in \{1, 2, \ldots, N\}$, and

- $J_j(\pi) > J_j(\pi')$ for at least one $j \in \{1, 2, \ldots, N\}$.

Here, $J_i(\pi)$ denotes the value of the $i$-th objective for the solution $\pi$. The above conditions imply that $\pi$ performs at least as well as $\pi'$ in all objectives and strictly better in at least one. Solutions that are not dominated by any other are termed *non-dominated* and collectively form the *Pareto front*.

**Definition 2.** (Pareto Optimality) Let $\mathcal{X}$ denote the set of feasible solutions, and let $J : \mathcal{X} \to \mathbb{R}^N$ be a vector-valued objective function where $J(\pi) = [J_1(\pi), J_2(\pi), \ldots, J_N(\pi)]^\top$ corresponds to the objective values associated with $\pi \in \mathcal{X}$. A solution $\pi^* \in \mathcal{X}$ is *Pareto optimal* if and only if no other solution $\pi' \in \mathcal{X}$ satisfies:

$$J_i(\pi') \geq J_i(\pi^*) \quad \forall i \in \{1, 2, \ldots, N\} \tag{22}$$

and

$$J_j(\pi') > J_j(\pi^*) \quad \text{for at least one } j \in \{1, 2, \ldots, N\}. \tag{23}$$

This ensures that $\pi^*$ is *non-dominated*, meaning that no other solution can improve one or more objectives without sacrificing performance in at least one other.

## I.2  ADVANTAGE OF POLICY IMPROVEMENT

Explicit policy improvement refers to methods that deliberately optimize at least one objective $J_i$, ensuring that the solution quality improves by maximizing one or more associated rewards $R_i$. This

approach is particularly crucial in designing multi-objective policies, as it guarantees measurable progress in one or more dimensions of performance.

### ADVANTAGE OF MOC COMPARED TO OTHER BASELINES

Our proposed method, **MOC**, explicitly optimizes all objectives while integrating controllability, ensuring a more balanced and efficient approach to policy improvement. In contrast:

- **Rewarded Soup** does not jointly optimize all objectives, which leads to suboptimal solutions.
- **RiC** focuses exclusively on controllability but lacks explicit mechanisms for policy improvement, limiting its ability to enhance solution quality.
- **MODPO** does not consider Pareto Optimality during training. Specifically, it trains $M$ separate LLMs (corresponding to $M$ preferences) by optimizing each model with a specific weighted combination of reward objectives, given the corresponding reward models.

By integrating both explicit policy improvement and controllability into a unified framework, **MOC** theoretically achieves higher solution quality compared to these baselines. This is further validated by our experimental results (Tables 1 to 4 and 10 to 12 and Figures 2, 3, 5 and 6), which demonstrate that **MOC** consistently outperforms these approaches across multiple metrics.

The integration of explicit policy improvement with controllability ensures that **MOC** aligns with the principles of Pareto Optimality while delivering superior practical performance. By addressing the limitations of existing methods and achieving a better balance among competing objectives, **MOC** sets a new benchmark in multi-objective controllable language models.

