# OpenReview forum: "One Model for All: Multi-Objective Controllable Language Models"
_ICLR.cc/2025/Conference — ICLR 2025 Conference Withdrawn Submission_

### Official Review · Reviewer_gxJD · 2024-10-29

**Soundness:** 3
**Presentation:** 3
**Contribution:** 2
**Rating:** 5
**Confidence:** 4

**Summary:**

This paper presents an approach for fine-tuning LLMs to adapt to diverse user preferences through a new Multi-Objective Control (MOC) algorithm. Unlike traditional methods, MOC requires only a single training pass and eliminates the need for human preference data. The authors introduce a preference vector that encodes various user preferences, which is then appended to the original prompt to guide the model’s response.

The optimization process involves two main steps: first, solving a min-norm problem (Equation 12) to determine dynamic weights that balance the objectives, followed by optimizing a scalarized objective (Equation 7) based on these weights. The authors highlight that MOC’s computational cost is on par with single-objective Reinforcement Learning with Human Feedback (RLHF), making it an efficient alternative. Experimental results demonstrate MOC's effectiveness across multiple criteria, including solution quality, controllability, diversity, and generalization.

**Strengths:**

1. Clear Motivation and Writing: The paper is clearly motivated and well-written, effectively framing the relevance of aligning LLMs with diverse user preferences.

2. Important and Practical Problem: The research problem—aligning LLMs to accommodate diverse user preferences—is both interesting and of high practical importance.

3. Efficient Optimization Approach: The authors introduce an upper bound on the original objective (Equation 8) to serve as an optimization surrogate, addressing the intractable nature of multi-objective optimization and enhancing computational efficiency.

**Weaknesses:**

1. Lack of Clarity in Formulas: The mathematical formulation is unclear, particularly in Equation 5, where the reward model $R^k$ is fixed. The sampled response $y$ does not incorporate the parameter $\theta$, so how to compute its gradient with respect to
$\theta$?

2. Marginal Improvement: The experimental results show that the Multi-Objective Control (MOC) does not significantly outperform other baselines in terms of solution quality, particularly in the Harmless & Helpful setting (as indicated in Figure 2(b)). The performance is also not clearly superior in the Humor & Helpful setting.

3. Limited novelty: The idea is similar to RiC (Yang et al., 2024).

4. Weak experimental design: The paper lacks a comparison with the MODPO method, which could provide valuable insights into the proposed method's effectiveness against a strong baseline. The experiments are limited to the Llama2-7B model. Testing the proposed method across a broader range of model sizes and types would strengthen the validity of the findings.

[1] Zeqiu Wu, et al. Fine-Grained Human Feedback Gives Better Rewards for Language Model Training. NeurIPS 2023.
[2] Zhilin Wang, et al. HelpSteer2: Open-Source Dataset for Training Top-Performing Reward Models. Arxiv 2024.
[3] Zhanhui Zhou, et al. Beyond One-Preference-Fits-All Alignment: Multi-Objective Direct Preference Optimization. ACL 2024.

**Questions:**

1. What do the numerical labels on the markers represent in Fig. 3 of the paper?

2. In Section 4.3, how do we obtain an unseen preference vector? Is it by randomly setting the values of $p_{i}$ in Eq. 3 of the paper?

3. Is there an SFT before PPO for the policy to understand input that contains both prompt and preference vectors?

---

> ### Author Response · Authors · 2024-11-20
> **Response to Reviewer gxJD 1/3**
>
> Dear Reviewer gxJD,
>
> We sincerely appreciate your positive feedback on the clarity of our writing, the importance of the problem, and the efficient optimization we proposed. Thanks for your thoughtful comments. Below, we address each concern in detail.
>
>
>
> > W1. Lack of Clarity in Formulas: The mathematical formulation is unclear, particularly in Equation 5, where the reward model $R^k$ is fixed. The sampled response $y$ does not incorporate the parameter $\theta$, so how to compute its gradient with respect to $\theta$?
>
> **A1:** We apologize for any confusion caused by the notation. In the paper, as explained in lines 144–145, $x$ represents the prompt/query, $y \sim \pi(x;\theta, \mathbf{p}^i)$ is LLM-generated response, and $\mathcal{D}$ is the prompt dataset. The reward vector $\mathbf{R}(x,y) = (R^1(x,y), R^2(x,y), \cdots, R^N(x,y))$ is associated with the $N$ optimization objectives $\\{J^i\\}_{i=1}^N$.
>
> **This setup is consistent with standard RLHF**. The response $y$ depends on the policy parameters $\theta$, which propagates through the sampling process $y \sim \pi(x; \theta, \mathbf{p}^i)$. This dependency allows the gradient of $R^k$ with respect to $\theta$ to be computed using standard policy-gradient techniques. To address this concern more explicitly, we have revised the manuscript to more clearly emphasize the dependency of $y$ on $\theta$ and to highlight its equivalence to standard RLHF practices.
>
>
>
>
>
>
> > W2. Marginal Improvement: The experimental results show that the Multi-Objective Control (MOC) does not significantly outperform other baselines in terms of solution quality, particularly in the Harmless & Helpful setting (as indicated in Figure 2(b)). The performance is also not clearly superior in the Humor & Helpful setting.
>
> **A2:** We respectfully argue that MOC provides significant and consistent advantages across key dimensions: controllability, solution quality, diversity, and generalization. Below, we provide additional evidence to support these claims:
>
> - **Controllability**: MOC excels in aligning model behavior with diverse preferences, as measured by the local order rate, which quantifies the proportion of adjacent data points maintaining a monotonic relationship with input preference vectors. As reported in **Table 2**, MOC achieves a local order rate of **0.889**, substantially outperforming baseline methods, all of which fail to preserve this order. This result highlights MOC’s superior ability to adapt to preference.
>
> - **Solution Quality**: MOC produces higher-quality solutions by converging more closely to the Pareto front, as assessed using the hyper-volume indicator, a standard metric for multi-objective optimization. Table 3 shows that MOC achieves a hyper-volume score of **10.91**, significantly surpassing the baselines (RiC: 7.97, MORLHF: 7.90, Rewarded Soups: 7.50). These results confirm that MOC produces high-quality solutions and balances multiple objectives effectively.
>
> - **Diversity of Solutions**: MOC generates a broader range of trade-offs, as reflected by higher entropy values. In **Table 4**, MOC achieves an entropy score of **1.76**, outperforming baselines (RiC: 1.51, MORLHF: 1.47, Rewarded Soups: 1.63). This underscores MOC’s ability to produce a more diverse solution set, catering to varying preferences.
>
> - **Generalization to unseen preferences**: MOC demonstrates robustness in generalizing to unseen preferences. In Figure 3, MOC's solutions dominate RiC's solutions in all cases. Furthermore, MOC consistently outperforms baseline methods under varying conditions, as quantified by hyper-volume (Table 5) and local order rate (Table 6). These results validate MOC’s adaptability and its robustness in handling diverse scenarios.

---

> ### Author Response · Authors · 2024-11-20
> **Response to Reviewer gxJD 2/3**
>
> > W3. Limited novelty: The idea is similar to RiC (Yang et al., 2024).
>
> **A3:** While both RiC and MOC aim to personalize LLMs, the following significant differences highlight the novelty and distinct contributions of MOC:
>
> - Formulation and Approach:
>     - MOC formulates controllability as a multi-objective policy optimization with preference-based constraints, solved via the proposed MOC algorithm.
>     - A key novelty is the surrogate problem (Equation 12), which reduces computational cost to near that of a single-objective PPO.
>     - In contrast, RiC relies on rejection sampling and lacks an explicit policy optimization framework, making its methodology fundamentally different.
>
>
> - Explicit Policy Optimization and Controllability:
>     - MOC explicitly optimizes the policy to align model behavior with user preferences, establishing a rigorous and systematic controllability framework.
>     - RiC does not perform explicit policy optimization, limiting its ability to maximize reward while aligning with preferences.
>
> - Performance Advantages:
>     - Thanks to its principled design, MOC significantly outperforms RiC in controllability, solution quality, diversity, and generalization, as substantiated by quantitative results in Tables 2–6.
>
> We summarize these key differences below for clarity:
>
> | Algorithm | Source of Controllability    | Explicit Policy Improvement? | Loss Function       |
> |-----------|------------------------------|------------------------------|---------------------|
> | **MOC**   | Multi-objective optimization with constraints| Yes                      | PPO-based           |
> | **RiC**   | Rejection sampling           | No                       | SFT-based           |
>
>
>
>
>
> > W4.1 Weak experimental design: The paper lacks a comparison with the MODPO method, which could provide valuable insights into the proposed method's effectiveness against a strong baseline.
>
> **A4.1** Thank you for highlighting the relevance of MODPO to our work. We have added a comparison with MODPO in Table 1 and included another strong baseline, MetaAligner (in NeurIPS 2024 [1]).
>
> However, it is important to emphasize that MOC focuses on controllability with two critical goals:
>
> - Training one LLM to generate personalized outputs for diverse user preferences.
> - Generalizing to unseen preferences with the once-trained LLM.
>
> In contrast, MODPO:
>
> - Trains M separate LLMs (where M corresponds to the number of preferences), targeting only a fixed set of pre-defined preferences.
> - Does not consider generalization to unseen preferences.
>
> These differences highlight the fundamental distinctions between MODPO's methodology and objectives and those of MOC.
>
>
>
> > W4.2 The experiments are limited to the Llama2-7B model. Testing the proposed method across a broader range of model sizes and types would strengthen the validity of the findings.
>
> **A4.2:** Thank you for suggesting an expansion of our evaluation. In response, we conducted additional experiments to address this concern:
>
> - New Dataset: We included the Reddit Summary dataset [2].
> - Additional Baseline: We compared against MetaAligner [1].
> - Model Variants and sizes: We tested MOC on Llama-3-8B [3].
>
> We have included these experiments in Appendix H of our revised paper.
>
> **Experiment: HH-RLHF**
>
> - Setup: Llama-3-8B, with MetaAligner added as a new baseline. MetaAligner seeks to optimize all rewards in this setting.
> - Results: As detailed in the table below, MOC-Llama3-8B dominates MetaAligner and other baselines in hyper-volume:
>
> | Model Name       | Hyper-volume |
> |-------------------|--------------|
> | MOC-Llama3-8B    | 10.435       |
> | MOC-Llama2-7B    | 9.513        |
> | RiC              | 9.257        |
> | MetaAligner      | 3.41         |
>
> - Visualization: The results are presented in this link: https://anonymous.4open.science/r/MOC-9E51/llama3_8b_helpful_harmless.svg , highlighting MOC's superior performance compared to other methods
>
>
>
>
> **Experiment: Reddit Summary**. Llama-3-8B with Summary (measure the quality of the summary) and Faithful (measures the faithfulness of the summary for the original post) reward models.
>
> - Results: MOC outperforms the baseline (RiC) significantly in terms of rewards:
>
>  Model Name       | Hyper-volume |
> |-------------------|--------------|
> | MOC-Llama3-8B    | 17.556       |
> | RiC              | 14.052        |
>
> - Visualization: https://anonymous.4open.science/r/MOC-9E51/RiC_Faithful_Summary.svg , showing MOC significantly outperforms baseline in terms of the rewards.

---

> > ### Author Response · Authors · 2024-11-20
> > **Response to Reviewer gxJD 3/3**
> >
> > >Q5 & Q6. What do the numerical labels on the markers represent in Fig. 3 of the paper? In Section 4.3, how do we obtain an unseen preference vector? Is it by randomly setting the values of $p_i$ in Eq. 3 of the paper?
> >
> > **A5 & A6:** The numerical labels in Figure 3 represent the preference weights for the reward associated with the x-axis. As described in Appendix F:
> >
> > - We uniformly sampled four distinct groups of random, unseen preference number $n$ from the range [1, 100].
> > - Each sampled number $n$ was normalized by dividing by 100 to compute the preference $p_1$ for the first reward.
> > - The second preference was calculated as $1-p_1$, ensuring the weights sum to one. Then the preference vector is $[p_1, p_2]$.
> >
> > For clarity, the numerical values $n$ are directly shown in Figure 3.
> >
> >
> >
> >
> > >Q7. Is there an SFT before PPO for the policy to understand input that contains both prompt and preference vectors?
> >
> > **A7:**
> >
> > - **No Preference Vector During Initial SFT:** As shown in appendix E, before the PPO step, the LLM undergoes a single round of SFT on the "chosen" dataset. This dataset does not include any preference vectors.
> >
> > - **Preference Vectors Added During RLHF:** The model learns to interpret and utilize preference vectors during the RLHF phase. During this stage, preference vectors are appended to prompts, allowing the model to interpret and adapt to varying user preferences.
> >
> >
> >
> > We hope these clarifications and additional experiments address your concerns and strengthen the contributions of our submission. Thank you again for your time and thoughtful feedback.
> >
> > Sincerely,
> >
> > The authors
> >
> >
> > [1] Yang K, Liu Z, Xie Q, et al. Metaaligner: Towards generalizable multi-objective alignment of language models[C]//The Thirty-eighth Annual Conference on Neural Information Processing Systems. 2024.
> >
> > [2] Stiennon N, Ouyang L, Wu J, et al. Learning to summarize with human feedback[J]. Advances in Neural Information Processing Systems, 2020, 33: 3008-3021.
> >
> > [3] https://huggingface.co/meta-llama/Meta-Llama-3-8B

---

> ### Comment · Reviewer_gxJD · 2024-11-25
>
> Thanks for the clarifications. After reading the response, I have decided to keep my rating.

---

> > ### Author Response · Authors · 2024-11-25
> >
> > Dear Reviewer gxJD,
> >
> > Thank you for your thoughtful feedback and for considering our responses. We understand that you have decided to maintain your rating, and we respect your perspective. We would like to ensure we fully address your concerns to improve our work.
> >
> > Could you kindly let us know if there are any remaining points you feel we have not addressed or clarified adequately? Your additional insights would be greatly appreciated.
> >
> > Thank you for your time and effort.
> >
> > Best regards,
> >
> > The authors

---

> > ### Author Response · Authors · 2024-12-02
> > **Final Reminder: Highlighting MODPO Comparison Results Before Discussion Deadline**
> >
> > Dear Reviewer gxJD,
> >
> > As the discussion period concludes in approximately 12 hours, we wanted to remind you of our responses, particularly the updated comparison with MODPO. The results, now included in Appendix H, demonstrate that MOC significantly outperforms MODPO and other baselines, especially in terms of hyper-volume (e.g., MOC-Llama3-8B achieves 10.435 compared to MODPO's 3.745).
> >
> > If there are any additional points you would like clarified or discussed further, please let us know before the discussion period ends.
> >
> > Best regards,
> >
> > The Authors

---

> ### Author Response · Authors · 2024-11-27
> **Comparison with MODPO**
>
> Dear Reviewer gxJD,
>
> We conducted additional experiments to include MODPO [1] as a baseline, and we updated the results in Appendix H. The new results demonstrated that our proposed method, MOC, significantly outperforms MODPO and other baselines, particularly in terms of hyper-volume:
>
>
> | Algorithm       | Hyper-volume |
> |-------------------|--------------|
> | MOC-Llama3-8B    | 10.435       |
> | MOC-Llama2-7B    | 9.513        |
> | RiC              | 9.257        |
> | MetaAligner      | 3.41         |
> | MODPO            | 3.745        |
>
>
> - Visualization: The results are presented in this link: https://anonymous.4open.science/r/MOC-9E51/llama3_8b_helpful_harmless.svg , highlighting MOC's superior performance compared to other methods
>
> - Setup: Llama-3-8B, with MODPO added as a new baseline. MODPO seeks to maximize all rewards in this setting.
>
>
> We are grateful for your feedback, as it allowed us to strengthen our comparisons and further validate the contributions of our method.
>
> Thank you again for your comments.
>
>
> Best wishes,
>
> The authors
>
>
> [1] Zhou Z, Liu J, Shao J, et al. Beyond one-preference-fits-all alignment: Multi-objective direct preference optimization[C]//Findings of the Association for Computational Linguistics ACL 2024. 2024: 10586-10613.

---

> ### Author Response · Authors · 2024-12-02
> **Reminder: Discussion Ends in Less Than 24 Hours**
>
> Dear Reviewer gxJD,
>
> We hope this message finds you well. As the rebuttal period concludes in less than 24 hours, we wanted to kindly remind you of our responses to your comments, particularly addressing the comparison with MODPO. In our revised paper, we have included additional experiments demonstrating that MOC significantly outperforms MODPO and other baselines.
>
> Your feedback has been invaluable in strengthening our work, and we are eager to address any remaining concerns you may have. If there are additional clarifications or points you need further clarify, please let us know.
>
> Thank you again for your time.
>
> Best regards,
>
> The Authors

---

### Official Review · Reviewer_BCRA · 2024-11-04

**Soundness:** 2
**Presentation:** 3
**Contribution:** 3
**Rating:** 6
**Confidence:** 3

**Summary:**

This paper tackles the challenge of aligning large language models (LLMs) with diverse human preferences without the impractical need to train separate models for each preference set. The authors introduce Multi-Objective Control (MOC), an algorithm that enables a single LLM to generate personalized outputs based on different user preferences along the Pareto front of multiple objectives. MOC integrates multi-objective optimization with Proximal Policy Optimization (PPO) which requires only one training phase without human preference data. The method shows that it is controllable, it offers diverse set of high quality solutions and it can handle unseen preferences during training.

**Strengths:**

1. The experiments include wide range of aspects. The authors evaluate its performance in terms of controllability, generalizability, Pareto front quality, and provide qualitative analyses.
2. Introducing preference vectors alongside multiple reward models allows the single policy model to adapt to different user preferences at inference time. This is an interesting idea in personalizing LLM outputs.
3. The paper effectively constructs the optimization problem, providing theoretical justifications and a clear algorithm.

**Weaknesses:**

1. The evaluation could be broadened by including more datasets and comparing MOC against additional baselines like Personalized Soups and MetaAligner.
2. The experiments are conducted solely with Llama 2 7B. Testing the approach on other models like Mistral or Gemma, and at different scales such as 13B, would strengthen the claim.
3. The method assumes the availability of N reward models or requires training N reward models, which can be computationally intensive.
4. The effectiveness of MOC isn't clearly illustrated in Figure 2b.

**Questions:**

1. Did you utilize pre-trained reward models, or did you train the reward models as part of your work?
2. How did you decide on the number of reward models N and the number of preference vectors M used in your experiments?

---

> ### Author Response · Authors · 2024-11-20
> **Response to Reviewer BCRA 1/3**
>
> Dear Reviewer BCRA,
>
> Thank you for your detailed and constructive feedback on our submission. Below, we provide clarifications, additional experiments, and evidence to address the concerns you raised.
>
>
> > W1&2: The evaluation could be broadened by including more datasets and comparing MOC against additional baselines like Personalized Soups and MetaAligner. The experiments are conducted solely with Llama 2 7B. Testing the approach on other models like Mistral or Gemma, and at different scales such as 13B, would strengthen the claim.
>
> **A1&2.**
>
> We appreciate your suggestion to expand the evaluation. In response, we have conducted additional experiments:
>
> - New Dataset: We included the Reddit Summary dataset [1].
> - Additional Baseline: We compared against MetaAligner [2].
> - Model Variants and sizes: We tested MOC on Llama-3-8B [7].
>
> These experiments are detailed in Appendix H of our revised paper.
>
> **Experiment: HH-RLHF**
>
> - Setup: Llama-3-8B, with MetaAligner added as a new baseline. MetaAligner aims to maximize all the rewards in this setting.
> - Results: As detailed in the table below, MOC-Llama3-8B dominates MetaAligner and other baselines in hyper-volume:
>
> | Model Name       | Hyper-volume |
> |-------------------|--------------|
> | MOC-Llama3-8B    | 10.435       |
> | MOC-Llama2-7B    | 9.513        |
> | RiC              | 9.257        |
> | MetaAligner      | 3.41         |
>
> - Visualization: The results are presented in this anonymous link: https://anonymous.4open.science/r/MOC-9E51/llama3_8b_helpful_harmless.svg , showing MOC's substantial performance compared to others.
>
>
>
>
> **Experiment: Reddit Summary**. Llama-3-8B with Summary (measure the quality of the summary) and Faithful (measures the faithfulness of the summary for the original post) reward models.
>
> - Results: MOC significantly surpasses the baseline (RiC) in rewards:
>
>  Model Name       | Hyper-volume |
> |-------------------|--------------|
> | MOC-Llama3-8B    | 17.556       |
> | RiC              | 14.052        |
>
> - Visualization: https://anonymous.4open.science/r/MOC-9E51/RiC_Faithful_Summary.svg , showing MOC significantly outperforms baseline in terms of the rewards.
>
>
>
> > W3: The method assumes the availability of N reward models or requires training N reward models, which can be computationally intensive
>
>
> **A3.** We appreciate your comment on the computational burden of $N$ reward models. Below, we outline why MOC is computationally efficient.
>
> - **Reward Models in RLHF:** Utilizing reward models is a standard paradigm in RLHF. These models are essential for capturing and quantifying human preferences. The use of $N$ reward models aligns with the RLHF framework.
>
> - **Minimal Overhead.** MOC requires training only a single model, unlike:
>      - **Rewarded Soup:** [3] requires training $N$ models, each fine-tuned on a single reward model, and then combines their weights. This approach incurs significantly higher computational costs.
>      - **MODPO:** [4] trains $M$ separate models (where $M$ is the number of user-defined preferences, i.e., 20 preference, 20 language models), further increasing the computational burden.
>      - **RiC:** [5] Conditions on multiple rewards but lacks explicit policy improvement, which can limit performance and necessitate additional computational adjustments of fine-tuning.
>    - Our method, in contrast, **trains a single model** that can generate to unseen preferences, minimizing training overhead while superior performance.
>
> - **Empirical Resource Efficiency:** Our method demonstrates exceptional resource efficiency:
>      - MOC allows fine-tuning a 7B parameter LLMs using **a single A6000 GPU**, as we did in the paper.
>      - Our experiments, as detailed in Sections 4.2 and 4.3, demonstrate superior performance with minimal resource consumption across hypervolume, alignment results, and generalization.

---

> ### Author Response · Authors · 2024-11-20
> **Response to Reviewer BCRA 2/3**
>
> > W4: The effectiveness of MOC isn't clearly illustrated in Figure 2b.
>
> **A4:** We appreciate your feedback regarding the clarity of MOC’s effectiveness in Figure 2b. Although Figure 2b may not immediately highlight MOC's advantages, Tables 2, 3, and 4 provide robust quantitative evidence demonstrating its effectiveness across three dimensions: controllability, solution quality, and diversity."
>
> 1. **Controllability:**
>
>     - Metric: We use the local order rate, which quantifies the proportion of adjacent data points maintaining a monotonic relationship with the input preference vectors.
>     - Results: As reported in **Table 2**, MOC achieves a local order rate of 0.778, significantly surpassing baseline methods, all of which fail to preserve order consistently.
>     - Visualization: As shown in Figure 2b, MOC’s solutions exhibit a smoother progression compared to the scattered behavior of the baselines.
>
>
> 2. **Solution Quality:**
>
>     - Metric: We evaluate solution quality using the hyper-volume indicator, a standard metric for assessing the quality of solution sets.
>     - Results: As shown in **Table 3**, MOC achieves a hyper-volume score of 9.51, outperforming RiC (9.26), MORLHF (9.05), and Rewarded Soups (8.90). This demonstrates MOC’s ability to produce high quality solutions that better balance the competing objectives.
>
>
> 3. **Diversity of Solutions:**
>
>     - Metric: We measure diversity using entropy, where higher values indicate a broader exploration of trade-offs across the Pareto front.
>     - Results: As presented in **Table 4**, MOC achieves an entropy score of 1.83, outperforming RiC (1.47), MORLHF (1.33), and Rewarded Soups (1.59). This highlights MOC's ability to effectively generate a more diverse set of solutions
>
>
> > Q5. Did you utilize pre-trained reward models, or did you train the reward models as part of your work?
>
> **A5.** To ensure consistency and reproducibility, we utilized publicly available pre-trained reward models. Table 9 in Appendix E provides details on the models used:
>
> | Reward            | Reward Model                                           | Link                                                                                                   |
> |--------------------|-------------------------------------------------------|--------------------------------------------------------------------------------------------------------|
> | Helpfulness       | gpt2 large helpful reward model                        | https://huggingface.co/Ray2333/gpt2-large-helpful-reward_model                                         |
> | Humor      | Humor no humor                                         | https://huggingface.co/mohameddhiab/humor-no-humor                                                     |
> | Harmlessness   | gpt2 large harmless reward model                       | https://huggingface.co/Ray2333/gpt2-large-harmless-reward_model                                        |
>
>
> Leveraging these pre-trained models ensured a robust and widely accepted evaluation framework.

---

> > ### Author Response · Authors · 2024-11-20
> > **Response to Reviewer BCRA 3/3**
> >
> > > Q6: How did you decide on the number of reward models N and the number of preference vectors M used in your experiments?
> >
> > **A6.** The selection of $N$ and $M$ was guided by the nature of the task and the need for comprehensive coverage of the objective space.
> >
> > - Number of Reward Models ($N$)
> >     - The number of reward models corresponds to the distinct objectives we sought to optimize, such as "Humor," "Helpfulness," and "Harmlessness."
> >     - This is consistent with prior work [2,3,4,5,6], where the objectives are derived directly from the dataset or application context.
> >
> > - Number of Preference Vectors ($M$)
> >     - Preference vectors were uniformly sampled ($p_1 \in \\{0, 0.1, \dots, 1.0\\}$) to ensure adequate coverage of the Pareto front. This is a general practice in literature [2,4,5].
> >     - This approach ensures that the model is exposed to diverse trade-offs between objectives. Furthermore, as demonstrated in Section 4.3, MOC generalizes effectively to unseen preferences, validating this design choice.
> >
> >
> >
> > We hope these clarifications and additional results address your concerns comprehensively. Please let us know if further explanations are needed. Thank you again for your thoughtful feedback.
> >
> > Best wishes,
> >
> > The authors
> >
> >
> > [1] Stiennon N, Ouyang L, Wu J, et al. Learning to summarize with human feedback[J]. Advances in Neural Information Processing Systems, 2020, 33: 3008-3021.
> >
> > [2] Yang K, Liu Z, Xie Q, et al. Metaaligner: Towards generalizable multi-objective alignment of language models[C]//The Thirty-eighth Annual Conference on Neural Information Processing Systems. 2024.
> >
> > [3] Rame A, Couairon G, Dancette C, et al. Rewarded soups: towards pareto-optimal alignment by interpolating weights fine-tuned on diverse rewards[J]. Advances in Neural Information Processing Systems, 2024, 36.
> >
> > [4] Zhou Z, Liu J, Shao J, et al. Beyond one-preference-fits-all alignment: Multi-objective direct preference optimization[C]//Findings of the Association for Computational Linguistics ACL 2024. 2024: 10586-10613.
> >
> > [5] Yang R, Pan X, Luo F, et al. Rewards-in-Context: Multi-objective Alignment of Foundation Models with Dynamic Preference Adjustment[C]//Forty-first International Conference on Machine Learning.
> >
> > [6] Jang J, Kim S, Lin B Y, et al. Personalized soups: Personalized large language model alignment via post-hoc parameter merging[J]. arXiv preprint arXiv:2310.11564, 2023.
> >
> > [7] https://huggingface.co/meta-llama/Meta-Llama-3-8B

---

> > > ### Author Response · Authors · 2024-11-25
> > > **Discussion Closing Soon: Your Feedback is Needed**
> > >
> > > Dear Reviewer BCRA
> > >
> > > We hope this comment finds you well. As the discussion phase for your reviewed submission will close in less than 48 hours, we kindly remind you to share any additional comments or questions you may have.
> > >
> > > We value your feedback and are committed to addressing your concerns promptly. Please let us know if there is anything specific you would like us to clarify or discuss further.
> > >
> > > Thank you for your time and effort in reviewing our work.
> > >
> > > Best regards,
> > >
> > > The authors

---

> ### Comment · Reviewer_BCRA · 2024-11-25
> **Response to the authors**
>
> Thank you for conducting the additional experiments and clarifying the questions. I will increase my rating.

---

> ### Author Response · Authors · 2024-11-25
>
> Dear Reviewer BCRA,
>
> Thank you for reviewing our clarifications and additional experiments :P We greatly appreciate your constructive feedback and your support to increase the rating. Your insights have been invaluable in improving our work.
>
> Best wishes,
>
> The authors

---

### Official Review · Reviewer_315N · 2024-11-06

**Soundness:** 2
**Presentation:** 3
**Contribution:** 2
**Rating:** 6
**Confidence:** 4

**Summary:**

This paper introduces MOC, a mutli-obejct optimization approach to align model's output with user preferences by balancing trade-offs across multiple objectives.

It involves a **preference vector** that represents user objectives, and use a dynamic scalarization technique that balances the objectives.
The authors conduct experiments on the Helpful Assistant dataset, the experimental results further demonstrate its superiority over baseline methods.

**Strengths:**

MOC offers significant computational efficiency and personalization flexibility, enabling single model to perform diverse preference-based tasks effectively

**Weaknesses:**

I still have concerns about the limitations in extreme preference scenarios, where trade-offs between objectives might not fully capture nuanced user expectations. Additionally, while the computational cost is comparable to single-objective PPO methods, more complex applications might still require higher computational resources.

**Questions:**

see weakness

---

> ### Author Response · Authors · 2024-11-20
> **Response to Reviewer 315N**
>
> Dear Reviewer 315N,
>
> Thank you for your thoughtful comments and for the time you dedicated to reviewing our paper. Below, we provide detailed responses to address your concerns.
>
>
> >W1: I still have concerns about the limitations in extreme preference scenarios, where trade-offs between objectives might not fully capture nuanced user expectations.
>
> **A1:** Thank you for raising this point regarding extreme preference scenarios. We would like to clarify how our approach addresses this concern:
>
> 1. **Preference Vector and Pareto Front Coverage:** The preference vector $\mathbf{p} = [p_1, p_2, \cdots, p_N]$, where $0 \leq p_i \leq 1$ and $\sum_{i} p_i = 1$, is explicitly designed to map user preferences across the Pareto front comprehensively. By representing multi-objective preference through weights $[w_1, w_2, \cdots, w_N]$ (where $0 \leq w_i \leq 1$), the preference vector ensures complete and flexible coverage of trade-offs. Consequently, our method effectively represents diverse user expectations within the defined preference space.
>
> 2. **Extreme Preferences and Human Judgment:** For scenarios involving extreme or highly nuanced preferences, additional refinement through human feedback is beneficial. This refinement requires precise data annotation during human preference collection, which is outside the primary scope of our MOC algorithm. Our approach focuses on controllability in multi-objective optimization rather than granular data annotation. As such, MOC provides a strong foundation for alignment with extreme preferences.
>
>
>
> >W2: Additionally, while the computational cost is comparable to single-objective PPO methods, more complex applications might still require higher computational resources.
>
> **A2:** We thank the reviewer for pointing out the potential computational considerations for complex applications. We address this concern from both theoretical and empirical perspectives:
>
> 1. **Theoretical Perspective:**
>    Single-objective PPO represents the **minimum** computational costs for PPO-based methods. Our MOC algorithm is designed to maintain this efficiency, even when addressing more complex scenarios. Specifically:
>    - MOC requires training only **one model** with **inference adaptation** and does not rely on **human preference data**.
>    - MOC enables explicit **policy improvement** and **policy controllability**, which distinguishes it from alternatives:
>      - MORLHF: Lacks theoretical guarantees, cannot support inference adaptation. Train $M$ LLMs. $M$ is the number of user preferences.
>      - Rewarded Soup: [1] Requires training $N$ models (where $N$ is the number of reward models), lacks explicit controllability.
>      - RiC: [2] Does not provide explicit policy improvement.
>    These advantages, as summarized in Table 1 of our paper, establish MOC as a computationally efficient alternative to other methods.
>
> 2. **Empirical Perspective:**
>    Experimentally, our results validate the resource efficiency of MOC:
>    - MOC achieves superior results while fine-tuning a single 7B parameter model on a **single A6000 GPU**. This approach is significantly more resource-efficient than baseline methods.
>    - Across all benchmarks, MOC demonstrates superior performance in terms of controllability, solution quality, diversity, and generalization to unseen preferences (see Sections 4.2 and 4.3)
>
> In conclusion, while it is expected that more complex applications may demand higher computational resources, MOC’s design ensures resource efficiency. Both theoretical guarantees and empirical results establish MOC as a highly scalable solution for multi-objective controllability, even in more complex scenarios.
>
> Thank you again for your valuable feedback. Please let us know if further clarification is required.
>
> Best regards,
>
> The authors
>
>
> [1] Rame A, Couairon G, Dancette C, et al. Rewarded soups: towards pareto-optimal alignment by interpolating weights fine-tuned on diverse rewards[J]. Advances in Neural Information Processing Systems, 2024, 36.
>
> [2] Yang R, Pan X, Luo F, et al. Rewards-in-Context: Multi-objective Alignment of Foundation Models with Dynamic Preference Adjustment[C]//Forty-first International Conference on Machine Learning.

---

> > ### Author Response · Authors · 2024-11-25
> > **Discussion Closing Soon: Your Feedback is Needed**
> >
> > Dear Reviewer 315N,
> >
> > We hope this comment finds you well. As the discussion phase for your reviewed submission will close in less than 48 hours, we kindly remind you to share any additional comments or questions you may have.
> >
> > We value your feedback and are committed to addressing your concerns promptly. Please let us know if there is anything specific you would like us to clarify or discuss further.
> >
> > Thank you for your time and effort in reviewing our work.
> >
> > Best regards,
> >
> > The authors

---

> > ### Comment · Reviewer_315N · 2024-12-02
> >
> > Thank you for your response. I have raised my score.

---

> > > ### Author Response · Authors · 2024-12-02
> > >
> > > Dear Reviewer 315N,
> > >
> > > Thank you for raising your score and for taking the time to engage with our responses. We greatly appreciate your recognition of our work and your thoughtful consideration.
> > >
> > > Best regards,
> > >
> > > The Authors

---

> ### Author Response · Authors · 2024-11-27
> **Follow-up Reminder**
>
> Dear Reviewer 315N,
>
> We hope this comment finds you well. We wanted to kindly remind you to review our rebuttal and share any additional comments or feedback you may have. Your input is highly valued, and we appreciate the time and effort you dedicate to this process.
>
> Thank you again for your consideration.
>
> Best regards,
>
> The Authors

---

> ### Author Response · Authors · 2024-11-28
> **Follow-up on Your Feedback**
>
> Dear Reviewer 315N,
>
> We hope this message finds you well. Thank you again for your earlier comments and insights. We have provided detailed responses and updates to your concerns and would greatly appreciate any further feedback or questions you might have before the discussion phase concludes.
>
> Your perspective is highly valued, and we are eager to address any remaining points to ensure the clarity and rigor of our work. Please don’t hesitate to reach out if additional clarifications are needed.
>
> Thank you for your time and consideration.
>
> Best regards,
>
> The Authors

---

> ### Author Response · Authors · 2024-12-02
> **Final Reminder: Discussion Deadline in Less Than 24 Hours**
>
> Dear Reviewer 315N,
>
> We hope this message finds you well. As the rebuttal period will end in less than 24 hours, we kindly remind you to review our earlier response to your valuable feedback. We have addressed your concerns regarding extreme preference scenarios and computational costs with detailed explanations.
>
> If there are any remaining questions or further clarifications you would like us to provide, we are fully committed to addressing them before the deadline. We deeply appreciate your time and effort.
>
> Thank you once again for your review and consideration.
>
> Best regards,
>
> The Authors

---

### Official Review · Reviewer_PAko · 2024-11-08

**Soundness:** 2
**Presentation:** 2
**Contribution:** 3
**Rating:** 5
**Confidence:** 2

**Summary:**

This paper studies training a single language to fit difference preferences. The paper formulate the problem as a multi-objective optimization problem. The formulated problem is then simplified to another optimization problem with less computational complexity. Experiment results on Helpful assistant show that MOC outperforms baseline methods.

**Strengths:**

The strengths of this paper are listed as follows

1. This paper studies an important problem, which is to fit a single model to diverse user preferences.

2. This paper models the problem as a multi-objective optimization problem. To tackle the challenge in the formulated problem, the authors use reward signal as target and also relax the complicated optimization problem to an easier one. These techniques are new to me.

3. The results of experiment looks good. The authors also include multiple dimension to evaluate the performance of MOC and generalization ability, which make the experiment comprehensive with respect to performances.

**Weaknesses:**

The weaknesses of the paper are listed below

1. The motivation of MOC are not clearly stated. However, this paper does not adequately discuss how previous methods view this problem and what the limitation of the previous approaches are. Therefore, the motivation behind formulating the problem as a MOO problem is not well-explained. Clarifying these shall make readers better understand the rationale of the method.

2. Some parts in method look confusing to me. For example, in equation (1), there seems a ambiguity regarding whether $J^i$ is a scaler or a vector. If it is a scalar, why there is a transpose ($\top$)? Otherwise, whats the definition of taking maximum over a bunch of vectors?Also, the intuition behind constraint in (1) could be better explained. Usually we want reward to be as large as possible, so why we want $R^i(x,y) \approx p_i$ even when $p_i$ is small?

3. The experiment is restricted to three objective (helpfulness, harmless, humor). However, in real world scenarios, users' demand might encompass much more objectives. Therefore, experiments on a larger number of objectives are required to show the generalization ability of MOC over different numbers of objectives.

**Questions:**

Can author provide some intuitive explainations on why the value of $c^{(1)},c^{(2)}$ is determined by (8)?

---

> ### Author Response · Authors · 2024-11-20
> **Response to Reviewer PAko 1/5**
>
> Dear Reviewer PAko,
>
> We sincerely thank the reviewer for the insightful comments and the recognition of the importance of the controllability, the novelty of our method, and the comprehensiveness of our evaluation.  Below, we address each of the concerns raised.
>
>
>
>
> > W1: The motivation of MOC are not clearly stated. However, this paper does not adequately discuss how previous methods view this problem and what the limitation of the previous approaches are. Therefore, the motivation behind formulating the problem as a MOO problem is not well-explained. Clarifying these shall make readers better understand the rationale of the method.
>
>
> **A1:** Thank you for highlighting the importance of better articulating the motivation and limitations of prior methods. Below, we elaborate on these aspects:
>
> 1. Motivation for MOC:
>     - Current LLMs are trained to align with fixed, developer-defined preferences, limiting their flexibility to address diverse user needs. This rigidity necessitates a paradigm that enables fine-grained controllability, enabling users to dynamically prioritize different objectives.
>     - Existing methods require either multiple specialized models or extensive preference datasets, both of which are resource-intensive and less scalable. In contrast, MOC formulates controllability as a multi-objective optimization (MOO) task, enabling dynamic preference-based trade-offs using a single model trained once.
>
>
> 2. Limitations of Prior Methods.
>     - Rewarded Soup [1] specializes $N$ LLMs independently (one per reward model) and then interpolates their weights linearly.
>     - MORLHF solve this problem by training individual $M$ LLMs with the the linearized rewards, where $M$ corresponds to the number of user-defined preferences.
>     - RiC [2] : Conditions LLM responses on multiple rewards using prompt conditioning, trained via rejection sampling.
>     - MODPO [3] trains $M$ LLMs, where $M$ corresponds to the number of preference, by optimizing each model with a specific weighted combination of reward objectives given the corresponding reward models.
>
> 3. Novelty of MOC:
>     - MOC uses a single LLM to dynamically adjust trade-offs among objectives based on user preferences.
>     - MOC explicitly improves policies while avoiding the need for multiple LLMs or preference datasets with a MOO problem formulation. This scalability and flexibility distinguish MOC from prior work.
>
>
>  **Comparison table**: To clarify MOC’s contribution, we summarize key differences with prior work in the table below:
>
>   | Algorithm                 | Explicit Policy Improvement | Number of Trained LLMs | Inference Adaptation | Preference Data | Loss Function |
> |---------------------------|-----------------------------|-------------------------|----------------------|-----------------|---------------|
> | **MORLHF**               | ✔                           | $M$                     | ✘                    | ✘               | PPO           |
> | **Rewarded Soups**        | ✘                           | $N$                     | ✔                    | ✘               | PPO           |
> | **MODPO**                 | ✔                           | $M$                     | ✘                    | ✔               | DPO           |
> | **RiC**                   | ✘                           | 1                       | ✔                    | ✘               | SFT           |
> | **MOC (Ours)**            | ✔                           | 1                       | ✔                    | ✘               | PPO           |

---

> ### Author Response · Authors · 2024-11-20
> **Response to Reviewer PAko 2/5**
>
> > W2: Some parts in method look confusing to me. For example, in equation (1), there seems a ambiguity regarding whether $J^i$ is a scaler or a vector. If it is a scalar, why there is a transpose ($^\top$)? Otherwise, whats the definition of taking maximum over a bunch of vectors? Also, the intuition behind constraint in (1) could be better explained. Usually we want reward to be as large as possible, so why we want $R^i(x,y) \approx p_i$ even when $p_i$ is small?
>
> **A2:**
>
>    - **$J^i$ is a Scalar:** Each $J^i(\pi(\cdot; \theta, \mathbf{p}^i))$ is a scalar representing the value of the $i$-th objective function.
>    - **Purpose of the Transpose$^\top$:** The notation $(J^1, J^2, \dots, J^N)^\top$ represents a **column vector**, ensuring compatibility with any potential matrix operations.
>    - **Maximizing in MOO:** In the context of MOO, "maximizing" the vector $\mathbf{J}$ refers to **finding Pareto optimal solutions**. A Pareto-optimal solution ensures no objective can be improved without deteriorating the others. This aligns with standard multi-objective optimization definitions.
>    - **Pareto Optimality:** The concept of Pareto optimality replaces the idea of a single global maximum in multi-objective contexts, providing a rigorous foundation for optimizing multiple competing objectives
>    - **Purpose of the Constraint:** The constraint ensures that the behavior of LLM (the rewards) **aligns** with user-preference** $\mathbf{p}^i$, acheving the controllability that MOC focus on. Maximizing reward is achieved through the optimization in equation 1: $\max_\theta \mathbf{J} (\pi (\cdot; \theta, \mathbf{p}^i)) := \max_\theta (J^1(\pi (\cdot; \theta, \mathbf{p}^i)), J^2(\pi (\cdot; \theta, \mathbf{p}^i)), \cdots, J^N(\pi (\cdot; \theta, \mathbf{p}^i)))^\top$. This constraint ensures alignment with user priorities.
>
> - **To ensure consistency in scale between the rewards and the preference vector, the rewards are normalized to align with the scale of the preference vector.** The reward signal is normalized by $r = \frac{r-r_\text{mean}}{2 r_{\text{std}}}+1$, as detailed in Appendix E, where the mean and std are computed by running mean in [4], which is a general practice in deep reinforcement learning [4]. This ensures consistency across rewards and preferences. We have updated Appendix G with additional clarifications.

---

> ### Author Response · Authors · 2024-11-20
> **Response to Reviewer PAko 3/5**
>
> > W3: The experiment is restricted to three objective (helpfulness, harmless, humor). However, in real world scenarios, users' demand might encompass much more objectives. Therefore, experiments on a larger number of objectives are required to show the generalization ability of MOC over different numbers of objectives.
>
> **A3:** We acknowledge the reviewer's concern regarding experiments with more objectives. While our primary experiments focus on three objectives, MOC can theoretically handle a larger number of objectives. To substantiate this claim, we conducted additional experiments using the 6-objective [Fruit-Tree task](https://mo-gymnasium.farama.org/environments/fruit-tree/) from the MO-Gymnasium benchmark. We have included these experiments in Appendix H of our revised paper.
>
> **1. Generalization to a Larger Number of Objectives**.
>
>
> Task Description: The Fruit-Tree task involves navigating a binary tree of depth 6, where each leaf contains a 6-dimensional reward vector corresponding to nutrient values. The action space is discrete, requiring the agent to optimize for all nutrients simultaneously. Additional details are available on the MO-Gymnasium website [5].
>
>
>
> The results (see Table below) demonstrate that MOC significantly outperforms the Linear PPO baseline in terms of hyper-volume, demonstrating its effectiveness in optimizing across multiple objectives.
>
>
> | Hyper-volume     | MOC       | Linear PPO    |
> |------------|-----------|-----------|
> | Mean       | 15605.90  | 5741.79   |
> | Variance   | 752.97    | 877.43    |
>
>
>
> We provide a visualization comparing MOC (warm color) to Linear PPO (cool color) in terms of the density distributions of objectives 1, 2, and 3: https://anonymous.4open.science/r/MOC-9E51/generalize-to-6-objective-fruit-tree.svg
>
> MOC’s solutions consistently dominate those of the baseline, underscoring its superior performance. Full implementation details and hyper-parameters are outlined below.
>
> |Setting |Value|
> |-----|--------|
> |RL backbone|PPO|
> | Number of random seeds| 5|
> |Discount ($\gamma$)|$0.99$       |
> |Optimizer  |Adam    |
> |Learning rate for networks |$3 × 10^{−4}$|
> |Number of hidden layers for all networks|$3$|
> |Number of hidden units per layer|$256$|
> |Activation function|ReLU|
> |Batch Size|$100$|
> |Gradient clipping |False|
> |Exploration method|Policy Entropy|
> |Entropy Coefficient|$0.001$|
> |epsilon-clip for PPO|$0.001$|
> |Epochs per PPO update|$3$|
> |Timesteps every update |$100$|
> |Maximum episode timesteps| $100$|
> |Number of episodes per preference sample |$20$|
> |Number of preference samples | $2400$ |
> |Evaluation episode| $10$|
>
>
>
>
> **2. Generalization to Different Reward Models**
>
> To evaluate MOC’s generalizability to various reward models, we applied it to the Reddit Summary task [6] using the Llama-3-8B model. The task involved two reward models:
>
> - Summary: Evaluates the quality of the generated summary.
> - Faithfulness: Assesses the factual consistency of the summary with the original text.
>
> The results:
> | Hyper-volume     | MOC       | RiC    |
> |------------|-----------|-----------|
> | Mean       | 17.556  | 14.052   |
>
> MOC achieves significantly better performance than the baseline across both reward dimensions. Visual results are provided:
>
> Performance of MOC: [https://anonymous.4open.science/r/MOC-9E51/MOC-Llama3-8B_Faithful_Summary.svg](https://anonymous.4open.science/r/MOC-9E51/MOC-Llama3-8B_Faithful_Summary.svg)
> Performance of baseline: [https://anonymous.4open.science/r/MOC-9E51/RiC_Faithful_Summary.svg](https://anonymous.4open.science/r/MOC-9E51/RiC_Faithful_Summary.svg)
>
>
>
>
> The additional experiments address your concerns by:
>
> - Demonstrating MOC’s capability to handle a larger number of objectives (6-objective Fruit-Tree task).
> - Showcasing MOC’s adaptability to different reward types (Reddit Summary task).
>
>
> We believe these results strengthen the case for MOC’s versatility and practical applicability.

---

> ### Author Response · Authors · 2024-11-20
> **Response to Reviewer PAko 4/5**
>
> > Q4: Can author provide some intuitive explainations on why the value of $c^{(1)}$ and $c^{(2)}$ is determined by (8)?
>
> **A4:** Yes!
>
> **Multi-task learning**
>
> In multi-task learning, the objective is to train a model that performs well across multiple tasks simultaneously. Each task $t$ is associated with its own loss function, $\hat{\mathcal{L}}^t(\theta^{sh}, \theta^t)$, where $\theta^{sh}$ represents parameters shared among all tasks, and $\theta^t$ are task-specific parameters.
>
>
> The shared parameters $\theta^{sh}$ are updated using gradients derived from each task: $g_t = \nabla_{\theta^{sh}} \hat{\mathcal{L}}^t(\theta^{sh}, \theta^t)$. These gradients often conflict because they may point in different directions, making it challenging to update $\theta^{sh}$ in a manner that benefits all tasks simultaneously.
>
>
> To address this challenge, the goal is to find a single update direction that:
>
>
> - Balances the contributions from all tasks.
> - Minimizes conflicts between tasks to ensure that the update does not harm any individual task.
>
>
> This is achieved through a weighted combination of the task gradients:
>
> $$
> G(c) = \sum_{t=1}^T c^{(t)} g_t= \sum_{t=1}^T c^{(t)} \nabla_{\theta^{sh}} \hat{\mathcal{L}}^t(\theta^{sh}, \theta^t),
> $$
>
> where $c^{(t)}$ are the weights determine each task's contribution. Note that i) $\sum_{t=1}^T c^{(t)} = 1$: Ensures that the weights form a convex combination, maintaining the update within the feasible region defined by the task gradients; and ii) $c^{(t)}t \geq 0$: This prevents reversing a task’s gradient direction, which could hinder optimization.
>
>
> **Equation (8) and Its Implications.**
>
> Equation (8) defines an optimization problem that minimizes the squared Euclidean norm of the weighted gradient combination:
>
> $$
> \min_{c^{(1)},\ldots,c^{(T)}} || G(c) ||_2^2.
> $$
>
> This optimization achieves two critical objectives:
>
> - **Conflict Reduction**: A smaller norm indicates less disagreement among task gradients. Minimizing this norm aligns the gradients into a more harmonious direction.
> - **Balanced Update**: The weights $c^{(t)}$ adaptively adjust the contribution of each task to ensure that no single task dominates the update direction unless necessary to reduce conflict.
>
>
> **Intuitive Explanation**
>
> We provide a geometric and intuitive understanding of this process. Check this link: https://anonymous.4open.science/r/MOC-9E51/min_norm_solution.svg for the figure.
>
> 1. Geometric Perspective:
>     - Consider each task gradient ($g_1$ and $g_2$) as vectors.
>     - The weighted sum $G(c)$ lies within the **convex hull** formed by the two gradient vectors.
>     - Minimizing $||G(c)||_2^2$ corresponds to finding the point in the convex hull that is closest to the origin. This point represents the most balanced update direction that aligns all task gradients.
>
> 2. Conflict Minimization: The $c^{(t)}$ obtained by solving Equation 8 de-emphasizes tasks with conflicting gradients and amplifies tasks with aligned gradients. This reduces overall gradient conflict.
>
> 3. Optimality. By finding a common direction with minimal conflict, the method ensures progress across all tasks without harming any single task, leading to optimality.
>
>
> 4. Why Are These Weights Optimal? The values of $c^{(t)}$ are determined by the optimization problem in equation 8 because it seeks the most balanced and agreeable update direction. Theoretically, as demonstrated by [7, 8], either: (i) The solution to this min-norm problem is zero, in which case the resulting point satisfies the KKT conditions, which is Pareto stationary; or (ii) The solution yields a gradient direction that improves all objectives.
>
>
>
> Thank you for your thoughtful feedback. We believe these clarifications and additional results strengthen the paper and its contributions.
>
> Best wishes,
>
> The authors

---

> ### Author Response · Authors · 2024-11-20
> **Response to Reviewer PAko 5/5**
>
> **References**
>
> [1] Rame A, Couairon G, Dancette C, et al. Rewarded soups: towards pareto-optimal alignment by interpolating weights fine-tuned on diverse rewards[J]. Advances in Neural Information Processing Systems, 2024, 36.
>
> [2] Yang R, Pan X, Luo F, et al. Rewards-in-Context: Multi-objective Alignment of Foundation Models with Dynamic Preference Adjustment[C]//Forty-first International Conference on Machine Learning.
>
> [3] Zhou Z, Liu J, Shao J, et al. Beyond one-preference-fits-all alignment: Multi-objective direct preference optimization[C]//Findings of the Association for Computational Linguistics ACL 2024. 2024: 10586-10613.
>
> [4] Prafulla Dhariwal, Christopher Hesse, Oleg Klimov, Alex Nichol, Matthias Plappert, Alec Radford, John Schulman, Szymon Sidor, Yuhuai Wu, and Peter Zhokhov. Openai baselines. https://github.com/openai/baselines, 2017.
>
> [5] https://mo-gymnasium.farama.org/environments/fruit-tree/
>
> [6] Stiennon N, Ouyang L, Wu J, et al. Learning to summarize with human feedback[J]. Advances in Neural Information Processing Systems, 2020, 33: 3008-3021.
>
>
> [7] Désidéri J A. Multiple-gradient descent algorithm (MGDA)[D]. INRIA, 2009.
>
> [8] Sener O, Koltun V. Multi-task learning as multi-objective optimization[J]. Advances in neural information processing systems, 2018, 31.

---

> > ### Comment · Reviewer_PAko · 2024-11-25
> >
> > Thank you for your response and I appreciate the effort the author is making to answer the questions. However, I think some of the concerns are still not well-addressed
> >
> > Q1: I appreciate the author making the table comparing different approaches. However, I think the baseline methods deserve more descriptions in the introduction. Currently, the introduction only provides a table but without any further illustration (e.g., why explicit policy improvement is favored). I admit that I am not familiar with this research direction. However, in my opinion, the topic is of broad interest and thus the papers shall be more followable to those who are not familiar with this specific area. Therefore I suggest the author to discuss the previous work more sufficiently.
> >
> > Q2: Thanks for the answer. I suggest the author to provide the definition of Pareto Optimal explicitly in the paper to make the paper more self-contained. The current version is still confusing to those not familiar to this area.
> >
> > Q3 & Q4: My concerns are mostly addressed and also thanks the author(s) for the elaborate answer.

---

> > > ### Author Response · Authors · 2024-11-25
> > >
> > > Dear Reviewer PAko,
> > >
> > >
> > > We sincerely appreciate your further feedback and have revised our manuscript to explicitly define Pareto Optimality and provide additional explanations to enhance the paper’s self-containment. The updated content has been added to Appendix I in the revised version of the paper.
> > >
> > > Below, we present a formal definition of Pareto Optimality, discuss its relevance to policy improvement, and compare our approach to existing baselines.
> > >
> > >
> > >
> > >
> > >
> > > **Formal Definition of Pareto Optimality**
> > >
> > > **Definition** (non-dominated): Let $\pi, \pi' \in \mathcal{X}$, where $\mathcal{X}$ is the set of feasible solutions. A solution $\pi$ is said to dominate another solution $\pi'$ if and only if:
> > > - $J_i(\pi) \geq J_i(\pi')$ for all $i \in {1, 2, \ldots, N}$, and
> > > - $J_j(\pi) > J_j(\pi')$ for at least one $j \in {1, 2, \ldots, N}$.
> > >
> > > Here, $J_i(\pi)$ denotes the value of the $i$-th objective for the solution $\pi$. The above conditions imply that $\pi$ performs at least as well as $\pi'$ in all objectives and strictly better in at least one. Solutions that are not dominated by any other are termed **non-dominated** and collectively form the Pareto front.
> > >
> > > **Definition** (Pareto Optimality): Let $\mathcal{X}$ denote the set of feasible solutions, and $J: \mathcal{X} \to \mathbb{R}^N$ be a vector-valued objective function where $J(\pi) = [J_1(\pi), J_2(\pi), \ldots, J_N(\pi)]^\top$ corresponds to the objective values associated with $\pi \in \mathcal{X}$. A solution $\pi^* \in \mathcal{X}$ is Pareto optimal if and only if no other solution $\pi' \in \mathcal{X}$ satisfies:
> > >
> > > $$
> > > J_i(\pi') \geq J_i(\pi^*) \quad \forall i \in \{1, 2, \ldots, N\}
> > > $$
> > >
> > > and
> > >
> > > $$
> > > J_j(\pi') > J_j(\pi^*) \quad \text{for at least one } j \in \{1, 2, \ldots, N\}.
> > > $$
> > >
> > > This ensures that $\pi^*$ is non-dominated, meaning that no other solution can improve one or more objectives without sacrificing performance in at least one other.
> > >
> > >
> > > **Advantage of Policy Improvement**
> > >
> > > Explicit policy improvement refers to methods that deliberately optimize at least one objective $J_i$, ensuring that the solution quality improves by maximizing one or more associated rewards $R_i$. This approach is particularly crucial in designing multi-objective policies, as it guarantees measurable progress in one or more dimensions of performance.
> > >
> > >
> > > **Advantage of MOC compared to other baselines**
> > >
> > >
> > > Our proposed method, MOC, explicitly optimizes all objectives while integrating controllability, ensuring a more balanced and efficient approach to policy improvement. In contrast:
> > >
> > > - Rewarded Soup does not jointly optimize all objectives, which leads to suboptimal solutions.
> > > - RiC focuses exclusively on controllability but lacks explicit mechanisms for policy improvement, limiting its ability to enhance solution quality.
> > > - MODPO does not consider Pareto Optimality during training. Specifically, it trains $M$ separate LLMs (corresponding to $M$ preferences) by optimizing each model with a specific weighted combination of reward objectives, given the corresponding reward models.
> > >
> > > By integrating both explicit policy improvement and controllability into a unified framework, MOC theoretically achieves higher solution quality compared to these baselines. This is further validated by our experimental results (Tables 1 to 4 and 10 to 12 and Figures 2, 3, 5 and 6), which demonstrate that MOC consistently outperforms these approaches across multiple metrics.
> > >
> > >
> > > The integration of explicit policy improvement with controllability ensures that MOC aligns with the principles of Pareto Optimality while delivering superior practical performance. By addressing the limitations of existing methods and achieving a better balance among competing objectives, MOC sets a new benchmark in multi-objective controllable language models.
> > >
> > >
> > > We hope these clarifications address your concerns. Please let us know if you have further questions.
> > >
> > > Best wishes,
> > >
> > > The authors

---

> > > ### Author Response · Authors · 2024-11-28
> > > **Gentle Reminder: Follow-up on Your Valuable Feedback**
> > >
> > > Dear Reviewer PAko,
> > >
> > > We hope this comment finds you well. Thank you again for your thoughtful comments and follow-up questions. We have provided detailed responses addressing your additional concerns, including formal definitions and further discussions about baseline methods.
> > >
> > > If you have any further feedback or questions, we would greatly appreciate your input. Your expertise is invaluable in refining our work, and we are eager to address any remaining points to ensure clarity and completeness.
> > >
> > > Thank you for your time and consideration.
> > >
> > > Best regards,
> > >
> > > The Authors

---

> > > ### Author Response · Authors · 2024-12-02
> > > **Final Reminder: Follow-Up on Your Feedback Before Discussion Deadline**
> > >
> > > Dear Reviewer PAko,
> > >
> > > We hope this message finds you well. As the discussion deadline is approaching (with approximately 12 hours remaining), we wanted to kindly remind you of our detailed responses to your valuable feedback. We have made revisions, including formal definitions, expanded discussions on baseline methods, and enhancements to the paper’s clarity, all guided by your constructive comments.
> > >
> > > We remain eager to address any additional concerns you might still have. If there are further points for clarification, we would greatly appreciate your feedback before the discussion period concludes.
> > >
> > > Thank you again for your time.
> > >
> > > Best regards,
> > >
> > > The Authors

---

> ### Author Response · Authors · 2024-12-02
> **Discussion Deadline in Less Than 24 Hours**
>
> Dear Reviewer PAko,
>
> We hope this message finds you well. As the discussion will **close** in less than 24 hours, we wanted to kindly remind you of our earlier response addressing your comments. We have provided detailed explanations, including formal definitions, expanded discussions on baseline methods, and revisions to enhance the paper’s clarity.
>
> Your feedback is invaluable to us, and we remain eager to address any remaining concerns you may have. If there are any additional points you would like us to clarify, we would greatly appreciate hearing from you before the rebuttal period concludes.
>
> Thank you again for your time and thoughtful review.
>
> Best regards,
>
> The Authors

---

### Official Review · Reviewer_QBHy · 2024-11-13

**Soundness:** 2
**Presentation:** 3
**Contribution:** 2
**Rating:** 6
**Confidence:** 2

**Summary:**

In contrast with aligning LLMs to general human preferences, this paper focused on training the LLMs is such a way that they are controllable and can output responses according to the specified preferences. Current methodologies in Reinforcement Learning from Human Feedback (RLHF) often hinge on static reward functions drawn from averaged human ratings. This static nature presents limitations in adapting to or controlling varying individual preferences. To tackle this, the authors propose a novel approach that expands beyond a single preference model to accommodate multiple objectives. Their solution is positioned against the traditional single-objective focus of RLHF, aiming to deliver a more adaptable model capable of directly reflecting individual preferences.

1. The problem formulation is insightful: framing the controllability problem as a MOO problem with preference being controls.
2. The paper introduces a surrogate objective that circumvents the computationally expensive operations typically required for direct MOO, such as multiple backpropagations.
3. Extensive experiments suggest that the proposed model consistently outperforms baseline RL methods by providing greater flexibility and generalizing across unseen preferences. This underscores the model's utility for real-world applications where preference distributions continually evolve.

**Strengths:**

1. By focusing on multi-objective control, the model allows for more nuanced alignments with human preferences, catering to a wider array of individual needs compared to single-objective models.

2. The introduction of a surrogate objective, which simplifies the computational demands typically associated with MOO, stands out as a significant advancement. This makes it feasible to implement more complex preference structures without prohibitive cost.

3. The ability of the model to generalize to new, unseen preference distributions highlights its potential for broader application and scalability in dynamic environments.

**Weaknesses:**

1. No variance is reported for any metric used in the evlaution which is a common thing in RL works.

2. The representation of the preference vector in the prompt is an important aspect of the paper and not explored at all in the paper. Does the preference vector need to be relative in magnitudes aka it just specifies an rank order of the preferences?  If that is the case, are their other representations of the vector possible and how does that impact the model? Can the model understand preference specified as decimals?

3. The method uses MSE loss between the preference vector and the rewards for each preference as reward for their MOO objective. The authors does not discuss any limitations or structure required for this to work. Should the reward and preference be of the same scale. What if the preference is specified as [p_1: 0.25, p_2: 30] and rewards are [r_1: 0.5, r_2: 40]

4. Some baseline approaches will help highlight the importance of the approach.  An interesting baseline can be to add preference vector in the prompt and define reward as MSE(p, r) and apply standard PPO/DPO methods.

**Questions:**

Please refer to the weakness section.

---

> ### Author Response · Authors · 2024-11-20
> **Response to Reviewer QBHy 1/2**
>
> Dear Reviewer QBHy:
>
> Thank you very much for your detailed review and insightful comments on our manuscript. We would like to address your concerns with the following clarifications.
>
>
> > W1: No variance is reported for any metric used in the evaluation which is a common thing in RL works.
>
> **A1**: We appreciate your highlighting the importance of variance reporting in RL settings.
>
> 1. **Variance as a measure of response diversity in our paper.** In our submission, Figure 3 reports variance to capture the diversity of responses across different prompts, reflecting generalization across varied preferences. This variance offers insights into the model’s adaptability to new settings.
>
> 2. **Relevance of variance in RL versus LLM settings.**
>
>     - In RL, variance is typically reported across multiple runs with different random seeds to assess an agent’s stability, consistency, and robustness to initialization or environmental stochasticity. Lower variance generally indicates better reliability.
>
>     - In LLMs, however, variance reflects response diversity across prompts rather than learning stability. Due to the pre-trained nature of LLMs, response-level variance provides **limited** diagnostic value for assessing training reliability.
>
>
> 3. **Focus on Controllability in our work.** Our primary focus is on multi-objective controllability—aligning the model’s outputs with diverse user preferences. This is best assessed by mean performance metrics and response alignment quality, as response-level variance is less informative for the objectives of our study.
>
>
>
> In conclusion, while variance is reported in Figure 3, our selected metrics more effectively capture the model’s controllability and alignment across preference configurations.
>
>
>
>
>
> > W2: The representation of the preference vector in the prompt is an important aspect of the paper and not explored at all in the paper. Does the preference vector need to be relative in magnitudes aka it just specifies an rank order of the preferences? If that is the case, are their other representations of the vector possible and how does that impact the model? Can the model understand preference specified as decimals?
>
> **A2:** Thank you for this insightful question. We clarify below:
>
>
> - Representation of the Preference Vector: In our work, the preference vector $\mathbf{p} = [p_1, p_2, \ldots, p_N]$ is normalized to satisfy $\sum_{i=1}^N p_i = 1$ and $p_i \geq 0$. Each $p_i$ represents the relative importance of the $i$-th objective, allowing the vector to be interpreted directly as weights for multi-objective optimization.
>
>
> - Incorporation into Prompts: The preference vector is incorporated into prompts using the format: Re-labeled prompt = <R1> $p_1$ <R2> $p_2$ ... <RN> $p_N$ {prompt}. This design ensures the model can directly utilize the vector as a part of the input for optimization.
>
> -  The model can understand preferences specified as decimals, as demonstrated in Section 4.3. Here, MOC generalizes to unseen preference vectors, showcasing its adaptability to a wide range of representations. This approach aligns with prior work, such as RiC [1], which also incorporates decimal-based relabeling for preference vectors.
>
>
>
> We hope this clarifies the design and representation choices for the preference vector.
>
>
>
> >W3: The method uses MSE loss between the preference vector and the rewards for each preference as reward for their MOO objective. The authors does not discuss any limitations or structure required for this to work. Should the reward and preference be of the same scale. What if the preference is specified as [p_1: 0.25, p_2: 30] and rewards are [r_1: 0.5, r_2: 40]
>
> **A3:** We agree that scale alignment between rewards and preference vectors is critical. In our work, we ensure this alignment using reward normalization, as detailed below:
>
> Reward Normalization: As described in Appendix E, we normalize the reward signal as: $r = \frac{r-r_\text{mean}}{2 r_{\text{std}}}+1$. Here, $r_{\text{mean}}$ and $r_{\text{std}}$ are computed using a running mean and standard deviation (as in [4]), ensuring that rewards align with the scale of the preference vector. This approach is a standard practice in RL [4] and ensures the reward and preference are of the same scale.
>
> We have included this analysis in Appendix G of our revised paper for a comprehensive understanding.

---

> > ### Author Response · Authors · 2024-11-20
> > **Response to Reviewer QBHy 2/2**
> >
> > >W4: Some baseline approaches will help highlight the importance of the approach. An interesting baseline can be to add preference vector in the prompt and define reward as MSE(p, r) and apply standard PPO/DPO methods.
> >
> >
> > **A4.**  We thank the reviewer for suggesting this baseline (denoted as IB-PPO) and provide the following response:
> >
> >
> > - **Limitations of Linearized Reward Structures**: IB-PPO, like other linearized reward methods (e.g., Linear PPO, MORLHF), reduces multi-objective optimization to a single-objective scalarization using fixed weights. Such methods often introduce bias, with one objective dominating others, as discussed in [2, 3]. This bias fundamentally restricts their ability to achieve fine-grained alignment with diverse preference vectors.
> >
> >
> > - **Experimental Comparison with IB-PPO**: We evaluated IB-PPO on the Fishwood task. The results (available in this anonymous link: https://anonymous.4open.science/r/MOC-9E51/IB-PPO_result.png ) show that IB-PPO consistently fails to align with preference vectors, demonstrating poor multi-objective controllability. These findings highlight the necessity of our MOC framework for addressing such tasks effectively.
> >
> >
> > - **Advantages of MOC over Linearized Baselines**: Unlike linearized reward structures, MOC explicitly handles non-linear interactions between objectives, enabling better alignment across diverse preference configurations. Our extensive experiments in Sections 4.2 and 4.3 demonstrate MOC’s superior performance in terms of solution quality, controllability, and preference alignment.
> >
> >
> > In conclusion, while IB-PPO is a valuable baseline, it lacks the nuanced multi-objective control required for our setting, reinforcing the importance of the proposed MOC framework.
> >
> > We hope this response addresses all concerns and further substantiates the contributions of our work. Thank you for your constructive feedback and for considering our response.
> >
> > Sincerely,
> >
> > The Authors
> >
> >
> > [1] Yang R, Pan X, Luo F, et al. Rewards-in-Context: Multi-objective Alignment of Foundation Models with Dynamic Preference Adjustment[C]//Forty-first International Conference on Machine Learning.
> >
> > [2] Stephen P Boyd and Lieven Vandenberghe. Convex optimization. Section 4.7. Cambridge university press, 2004.
> >
> > [3] Agrawal T. Hyperparameter optimization in machine learning: make your machine learning and deep learning models more efficient[M]. New York, NY, USA:: Apress, 2021.
> >
> > [4] Prafulla Dhariwal, Christopher Hesse, Oleg Klimov, Alex Nichol, Matthias Plappert, Alec Radford, John Schulman, Szymon Sidor, Yuhuai Wu, and Peter Zhokhov. Openai baselines. https://github.com/openai/baselines, 2017.

---

> > > ### Author Response · Authors · 2024-11-25
> > > **Discussion Closing Soon: Your Feedback is Needed**
> > >
> > > Dear Reviewer QBHy,
> > >
> > > We hope this comment finds you well. As the discussion phase for your reviewed submission will close in less than 48 hours, we kindly remind you to share any additional comments or questions you may have.
> > >
> > > We value your feedback and are committed to addressing your concerns promptly. Please let us know if there is anything specific you would like us to clarify or discuss further.
> > >
> > > Thank you for your time and effort in reviewing our work.
> > >
> > > Best regards,
> > >
> > > The authors

---

> ### Author Response · Authors · 2024-11-27
> **Follow-up Reminder**
>
> Dear Reviewer QBHy,
>
>
> We hope this comment finds you well. We wanted to kindly remind you to review our rebuttal and share any additional comments or feedback you may have. Your input is highly valued, and we appreciate the time and effort you dedicate to this process.
>
> Thank you again for your consideration.
>
> Best regards,
>
> The Authors

---

> ### Author Response · Authors · 2024-11-28
> **Gentle Reminder: Your Feedback is Highly Valued**
>
> Dear Reviewer QBHy,
>
> We hope this comment finds you well. As the discussion phase is nearing its conclusion, we kindly reach out again to request your input on our rebuttal. We value your expertise and greatly appreciate your feedback, which is essential to improving the quality of our work.
>
> If you have any remaining questions or require clarification on any aspect of the submission, please let us know. We are happy to address your concerns promptly.
>
> Thank you for your time and for contributing to the review process.
>
> Best regards,
>
> The Authors

---

> ### Comment · Reviewer_QBHy · 2024-12-01
> **Response to the rebuttal**
>
> I would like to thanks the authors and acknowledge that I have read their response to the all the reviews.

---

> > ### Author Response · Authors · 2024-12-01
> >
> > Dear Reviewer QBHy,
> >
> > Thank you for taking the time to review our response and for acknowledging our efforts. We appreciate your consideration and the updated rating.
> >
> > Best regards,
> >
> > The authors

---

### Author Response · Authors · 2024-11-20
**General Response**

Dear Reviewers and Area Chair,

We sincerely thank you for your thoughtful reviews, constructive feedback, and positive remarks on our manuscript. We greatly appreciate your recognition of the importance of controllability, the clarity of our writing, the soundness of our methodology, and the efficiency of our proposed method, MOC.

We have carefully addressed each of your comments and concerns in our detailed responses. Your valuable insights have been instrumental in refining and enhancing the quality of our work. Additionally, we appreciate the Area Chair's efforts in coordinating in the review process. To assist you in identifying the revisions, we used $\color{teal}{\text{teal}}$ to highlight the updated content in the manuscript.


Sincerely,

The Authors

---

### Author Response · Authors · 2024-11-24
**Request for Response to Rebuttal for Your Reviewed Paper: One Model for All: Multi-Objective Controllable Language Models**

Dear Reviewers,

We hope this comment finds you well. We are writing to **kindly request your response** to the rebuttal of your reviewed paper: One Model for All: Multi-Objective Controllable Language Models.

We sincerely appreciate the time and effort you have dedicated to providing detailed feedback on our paper. Following the review phase, we carefully prepared a comprehensive rebuttal addressing each of the points you raised. To aid your review, the updated content is clearly marked in $\color{teal}{\text{teal}}$ in the revised paper.

However, we have not yet received any response or follow-up regarding our rebuttal.


As the discussion phase is approaching its conclusion, we would greatly value your input regarding the rebuttal. We fully understand the demands of the review process and deeply value your time, insights, and expertise.  If there are specific aspects of the rebuttal or revised paper that you would like us to elaborate on, please do not hesitate to let us know. Your feedback ensures a constructive and thorough review process.

Thank you very much for your time and attention. We look forward to hearing from you.

Best regards,

The Authors

---

### Note · Authors · 2025-01-22

I have read and agree with the venue's withdrawal policy on behalf of myself and my co-authors.